# Quantitative input–output dynamics of a c-di-GMP signal transduction cascade in *Vibrio cholerae*

**Andrew A. Bridges**[1○], **Jojo A. Prentice**[1○], **Chenyi Fei**[1,2], **Ned S. Wingreen**[1,2]*, **Bonnie L. Bassler**[1,3]*

**1** Department of Molecular Biology, Princeton University, Princeton, New Jersey, United States of America, **2** Lewis-Sigler Institute for Integrative Genomics, Princeton University, Princeton, New Jersey, United States of America, **3** The Howard Hughes Medical Institute, Chevy Chase, Maryland, United States of America

○ These authors contributed equally to this work.

* wingreen@princeton.edu (NSW); bbassler@princeton.edu (BLB)

## Abstract

Bacterial biofilms are multicellular communities that collectively overcome environmental threats and clinical treatments. To regulate the biofilm lifecycle, bacteria commonly transduce sensory information via the second messenger molecule cyclic diguanylate (c-di-GMP). Using experimental and modeling approaches, we quantitatively capture c-di-GMP signal transmission via the bifunctional polyamine receptor NspS-MbaA, from ligand binding to output, in the pathogen *Vibrio cholerae*. Upon binding of norspermidine or spermidine, NspS-MbaA synthesizes or degrades c-di-GMP, respectively, which, in turn, drives alterations specifically to biofilm gene expression. A long-standing question is how output specificity is achieved via c-di-GMP, a diffusible molecule that regulates dozens of effectors. We show that NspS-MbaA signals locally to specific effectors, sensitizing *V. cholerae* to polyamines. However, local signaling is not required for specificity, as changes to global cytoplasmic c-di-GMP levels can selectively regulate biofilm genes. This work establishes the input–output dynamics underlying c-di-GMP signaling, which could be useful for developing bacterial manipulation strategies.

## Introduction

Bacteria have the versatility to modify their lifestyles in response to shifting challenges. Commonly, bacteria resist threats by forming multicellular communities called biofilms, in which bacteria attach to surfaces and collectively produce an extracellular matrix [1]. Advantages that accrue to biofilm-resident cells include resistance to antimicrobial compounds, protection from predators, and the ability to collectively acquire nutrients [2–4]. Biofilm bacteria can return to the individual, free-swimming state by degrading the biofilm matrix and initiating motility, which together facilitate escape from the biofilm and spread to new territories [5]. The ability to repeatedly transition between planktonic and biofilm states is central to the disease process for many pathogenic bacteria, and such lifestyle flexibility confounds clinical

**Data Availability Statement:** All data, code, and the model derivation have been uploaded as main or supplemental files in this submission, or are

available on Zenodo (https://zenodo.org/record/5519935).

**Funding:** This work was supported by the Howard Hughes Medical Institute (B.L.B.); the National Science Foundation through the Center for the Physics of Biological Function PHY-1734030 (N.S. W.), as well as NSF grants MCB-2043238 (B.L.B.) and MCB-1853602 (B.L.B. and N.S.W.); NIH grants 1R21AI146223 (B.L.B. and N.S.W.), 2R37GM065859 (B.L.B.), GM082938 (N.S.W.), and 1K99AI158939 (A.A.B); and the Max Planck Society-Alexander von Humboldt Foundation (B.L. B.). During this work, A.A.B. was a Howard Hughes Medical Institute Fellow of the Damon Runyon Cancer Research Foundation (DRG-2302-17). The funders had no role in study design, data collection and analysis, decision to publish, or preparation of the manuscript.

**Competing interests:** The authors have declared that no competing interests exist.

**Abbreviations:** c-di-GMP, cyclic diguanylate; cpm, counts per million; ITC, isothermal titration calorimetry; LB, lysogeny broth; MIGS, Microbial Genome Sequencing Center; qlfTest, Quasi-Linear F-Test; TMM, Trimmed Mean of M values; *vps*, vibrio polysaccharide biosynthesis genes.

treatment [5]. Interventions that manipulate biofilm formation and/or dispersal hold promise as therapeutics to combat globally important pathogenic bacteria [6].

Biofilm formation and dispersal are controlled by environmental stimuli. Bacteria detect many of these stimuli with membrane-bound receptors that transduce signals internally. Information flow typically converges on a set of key intracellular regulators, commonly including the bacterial second messenger molecule cyclic diguanylate (c-di-GMP) [7]. Across the bacterial domain, high levels of c-di-GMP are associated with surface attachment and biofilm matrix production, while low levels drive inhibition of biofilm formation and enhanced motility [8]. Receptors controlling c-di-GMP levels contain ligand binding domains, which regulate the activities of attached domains responsible for c-di-GMP synthesis and/or degradation. c-di-GMP synthesis occurs via diguanylate cyclase domains, which harbor catalytic "GGDEF" motifs. c-di-GMP is degraded by phosphodiesterase domains characterized by either "EAL" or "HD-GYP" motifs [9,10]. The outputs of these enzymes are decoded by c-di-GMP–binding effectors that modulate transcription, translation, or protein activity, which, in turn, control bacterial behaviors [7]. For example, in the global pathogen *Vibrio cholerae*, the model organism used in the present work, the VpsT and VpsR transcription factors bind to c-di-GMP and subsequently activate expression of vibrio polysaccharide biosynthesis genes (*vps*) [11]. Biosynthesis of the Vps matrix promotes biofilm formation.

Bacterial genomes often encode dozens of c-di-GMP metabolizing receptors [12]. *V. cholerae* possesses 31 diguanylate cyclases, 12 phosphodiesterases, and 10 proteins containing both diguanylate cyclase and phosphodiesterase motifs [13]. This large array of receptors presumably allows *V. cholerae* to respond to many cues. However, it is mysterious how the pertinent sensory information is integrated to accurately regulate downstream processes including biofilm formation and dispersal. In *V. cholerae* and most other studied species, the ligands regulating specific receptors that harbor c-di-GMP synthesis and/or degradation domains remain unidentified [7]. Thus, a quantitative understanding of the input–output dynamics underlying how c-di-GMP signaling modules convert ligand binding events into alterations in phenotypes remains elusive. Furthermore, despite the diffusibility of the c-di-GMP molecule, genetic evidence suggests that some c-di-GMP–metabolizing enzymes show specificity, controlling only small subsets of c-di-GMP–binding effectors [12,14]. For example, deletion of genes specifying particular diguanylate cyclases reduces biofilm formation while not altering motility, whereas deletion of others activates motility but does not change biofilm formation [15]. How specificity is achieved by c-di-GMP signaling modules remains a topic of intense interest [12,14].

Here, we analyze the input–output dynamics of c-di-GMP signaling by MbaA, an inner membrane polyamine receptor that we previously showed controls biofilm dispersal in *V. cholerae* [16]. MbaA is a bifunctional enzyme, harboring both diguanylate cyclase (GGDEF domain) and phosphodiesterase (EAL domain) activities (Fig 1A). MbaA detects polyamine ligands via interaction with the periplasmic binding protein NspS [17,18]. NspS binds both norspermidine, a rare polyamine produced by *Vibrionaceae* and select other organisms, and spermidine, a nearly ubiquitous polyamine that is not produced by *V. cholerae* (Fig 1A) [19]. Norspermidine binding to NspS is thought to drive NspS-MbaA association, and this interaction favors MbaA diguanylate cyclase activity, promoting the biofilm state [19]. Conversely, when NspS is in the apo state or when it is bound to spermidine, NspS does not associate with MbaA, and MbaA exhibits phosphodiesterase activity, repressing biofilm formation and driving the planktonic state (Fig 1A) [19]. Thus, NspS ligand occupancy dictates MbaA activity, which, in turn, controls cytoplasmic c-di-GMP levels and whether *V. cholerae* forms or exits from biofilms. Because norspermidine is a rare polyamine in the biosphere whereas spermidine is widely produced across domains (and is present in the mammalian intestine at micromolar concentrations), the hypothesis is that the two polyamines allow *V. cholerae* to decipher

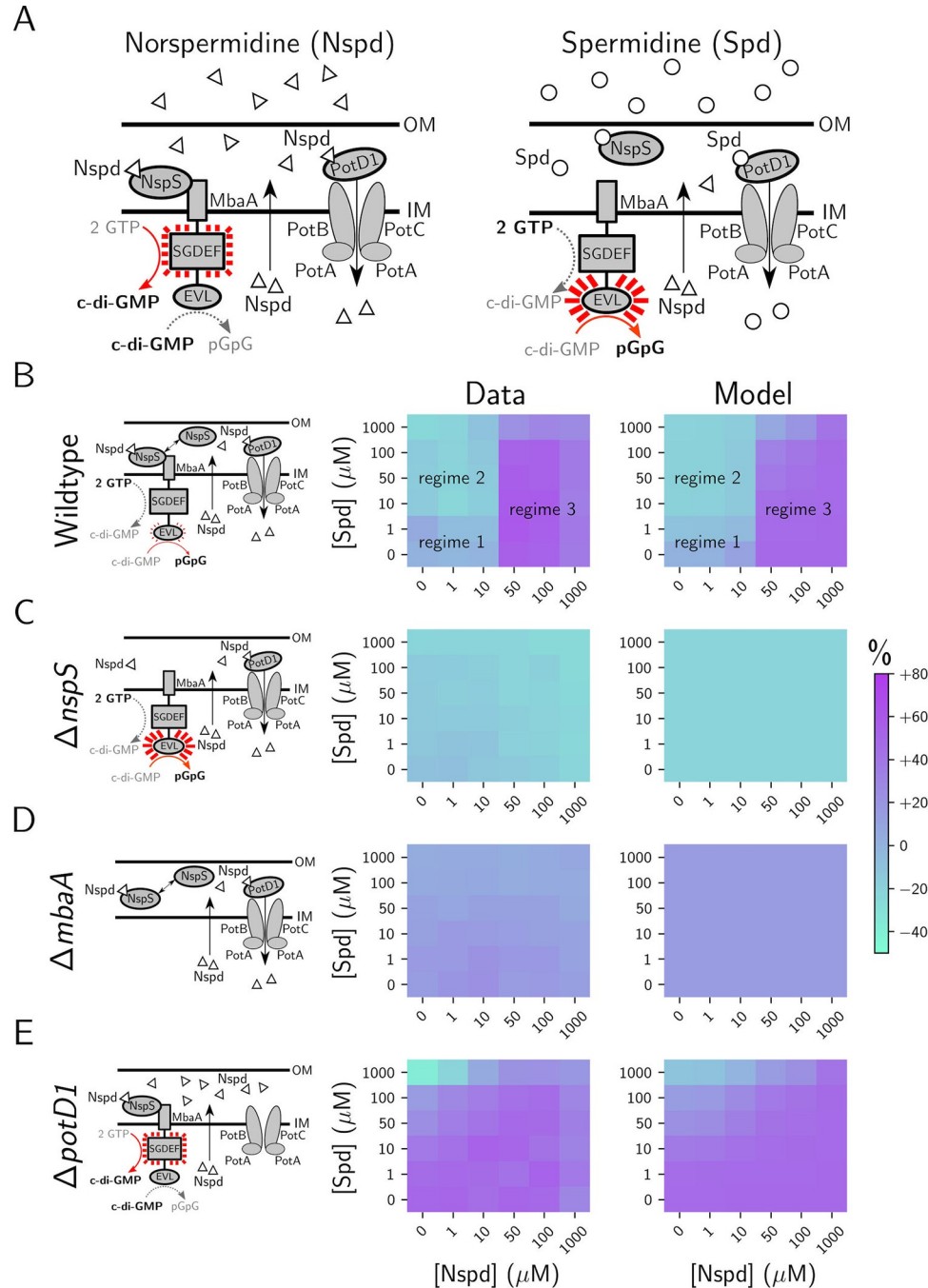

**Fig 1. Modeling of polyamine sources, transport, and detection by the NspS-MbaA circuit captures in vivo dynamics.** (A) Scheme for the NspS-MbaA c-di-GMP polyamine signal transduction circuit. *V. cholerae* detects periplasmic Nspd and Spd via the NspS-MbaA signaling system. Left panel: Detection of Nspd by the periplasmic binding protein NspS drives its association with MbaA, promoting MbaA diguanylate cyclase activity and biofilm formation. Nspd is exported to the *V. cholerae* periplasm via an unknown mechanism and is reimported by the PotABCD1 system. Right panel: When Nspd is absent, or when Spd is detected, NspS dissociates from MbaA and MbaA functions as a c-di-GMP phosphodiesterase, biofilm formation is repressed, and biofilm dispersal is promoted. Nspd is produced by *Vibrionaceae*. Spd is a commonly produced polyamine but is not substantially produced by *V. cholerae*. In MbaA, the GGDEF and EAL domains have the sequences SGDEF and EVL, respectively. MbaA was previously shown to possess both diguanylate cyclase and phosphodiesterase activities [16]. (B) Left panel: schematic of the periplasmic polyamine sensing and import components in wild-type *V. cholerae* in the absence of exogenous polyamines. Middle panel: experimentally obtained results for c-di-GMP reporter output in wild-type *V. cholerae* for

the indicated polyamine concentrations, displayed as a heatmap. Throughout the manuscript, data in c-di-GMP output heatmaps are displayed as percent differences compared to the untreated wild-type strain, with teal representing low and purple representing high c-di-GMP reporter output, respectively. Right panel: modeled c-di-GMP reporter output for wild-type *V. cholerae* based on a global fit to the experimental data for the wild-type, Δ*nspS*, Δ*mbaA*, and Δ*potD1* strains. (C) As in B for the Δ*nspS* strain. (D) As in B for the Δ*mbaA* strain. (E) As in B for the Δ*potD1* strain. *N* = 3 biological replicates. Numerical values for plots are available in S1 Data. The experimental results shown in the middle panels in B-E are reproduced from Bridges and Bassler, published under a Creative Commons Attribution license: https://creativecommons.org/licenses/by/4.0/. c-di-GMP, cyclic diguanylate; IM, inner membrane; Nspd, norspermidine; OM, outer membrane; Spd, spermidine.

the relatedness of other bacterial species in the vicinal community [16,18,20–22]. When closely related species are detected (via norspermidine), the biofilm lifestyle is favored, whereas when nonrelated species are detected (via spermidine), dispersal occurs and *V. cholerae* commits to the planktonic state, presumably to flee competitors. Notably, because the ligands for NspS-MbaA are known, and because MbaA can both produce and degrade c-di-GMP, this system is uniquely configured for a case study that incorporates all steps in c-di-GMP signal transduction—from ligand binding to phenotypic output.

To develop a quantitative understanding of signal transduction through the NspS-MbaA circuit, we formulate a mathematical model describing NspS-MbaA signaling from ligand binding to changes in c-di-GMP levels. Companion experiments show that NspS-MbaA detects its ligands in the periplasm with sub-nanomolar affinity. This sensitivity enables NspS-MbaA to respond to external polyamine fluctuations in the face of high affinity polyamine import by the cell. In turn, the c-di-GMP produced or degraded by MbaA regulates biofilm gene expression with high specificity. We address the long-standing issue of whether c-di-GMP signaling specificity is a consequence of localized c-di-GMP transmission between specific pairs of receptors and effectors and/or if changes in the global cytoplasmic c-di-GMP pool are detected by particular effectors based strictly on their relative affinities for c-di-GMP [14]. We find that MbaA appears to channel c-di-GMP information exclusively through a localized mechanism at low ligand concentrations, whereas at high ligand concentrations, it alters the global cytoplasmic c-di-GMP pool. Notably, however, across the large range of tested polyamine concentrations, MbaA maintains signaling specificity, as among all c-di-GMP–responsive genes, changes only occur in the expression of those involved in biofilm formation. By assaying other receptors that produce c-di-GMP, we show that localized c-di-GMP signaling is not a prerequisite for specific control of biofilm gene expression in *V. cholerae*, as specificity can also be achieved through changes to the global cytoplasmic c-di-GMP pool. Regarding the NspS-MbaA circuit, the consequence of local signaling is increased sensitivity to polyamine ligands, endowing *V. cholerae* with the ability to detect physiological polyamine concentrations. To our knowledge, the work presented here provides the first quantitative study of c-di-GMP signal transmission from ligand binding to changes in the behaviors of cell collectives.

## Results

### A mathematical model describes the experimentally observed NspS-MbaA input–output dynamics

To quantitatively describe the relationship between polyamine signal input and c-di-GMP output via the NspS-MbaA circuit, we relied on our previous experimental measurements that exploited an established fluorescence-based live cell c-di-GMP reporter [15,16,23]. In these assays, exogenous norspermidine and spermidine mixtures were supplied to cultures harboring a reporter whose output tracks linearly with cytoplasmic c-di-GMP concentrations [23]. The experimentally obtained wild-type c-di-GMP reporter results exhibit three regimes (Fig 1B,

left and middle panels): (1) a basal level of c-di-GMP, which is maintained when up to 10 μM exogenous norspermidine and up to 1 μM exogenous spermidine are added; (2) a reduced c-di-GMP level, which is established when high ($\geq$10 μM) spermidine and low ($\leq$10 μM) norspermidine are provided; and (3) an elevated c-di-GMP level, which is achieved when high ($\geq$50 μM) norspermidine is supplied, irrespective of the amount of administered spermidine. In the Δ*nspS* strain (Fig 1C, left and middle panels), basal c-di-GMP levels are lower than the wild-type basal level due to constitutive MbaA phosphodiesterase activity, and there is no response to exogenous polyamine mixtures. The Δ*mbaA* strain (Fig 1D, left and middle panels) is also incapable of responding to exogenous polyamines; however, it harbors slightly higher c-di-GMP levels than the untreated wild-type strain due to the lack of basal MbaA phosphodiesterase activity. The final experimental dataset we employed for model construction was that of the Δ*potD1* strain (Fig 1E, left and middle panels), which is incapable of norspermidine and spermidine import due to the lack of a functional PotABCD1 ABC-family transporter [19,24]. In the absence of polyamine import, our previous results suggest that self-secreted norspermidine accumulates in the periplasm, leading to high MbaA diguanylate cyclase activity (Fig 1E, left panel). Therefore, the Δ*potD1* strain exhibits elevated basal c-di-GMP levels relative to the wild type, no response to exogenous norspermidine, and an attenuated response to exogenous spermidine (Fig 1E, middle panel).

To uncover the features of the NspS-MbaA signaling circuit that underpin the measured dose responses, we designed a free-energy model describing the relationship between periplasmic polyamine concentrations and MbaA enzymatic activity. In free-energy models, protein configurations, and, therefore, activities, are drawn from the Boltzmann distribution. Depending on the free-energies associated with each configuration, which are fit to data, a population of two-state proteins may therefore almost entirely exist in one configuration or the other, or the population may be divided between the two states. In the model, we propose that norspermidine and spermidine influence MbaA activity exclusively through their effects on the chemical equilibrium among NspS conformations. Specifically, we assume that NspS can exist in one of two conformations: an "open" conformation, which is favored by spermidine binding but which does not interact with MbaA, and a "closed" conformation, which is favored by norspermidine binding and which drives the NspS-MbaA interaction. Notably, in the absence of polyamine binding, an intrinsic free-energy offset between the NspS open and closed conformations determines the equilibrium abundances of the two protein conformations. A given MbaA receptor, in turn, can exist in any of three states: unbound to NspS and exhibiting phosphodiesterase activity, unbound to NspS and exhibiting diguanylate cyclase activity, or bound to the closed conformation of NspS and exhibiting diguanylate cyclase activity. The chemical equilibrium among the MbaA states determines the average activity across all MbaA receptors and is calculated as a function of the free-energy offset between the phosphodiesterase and diguanylate cyclase states, $f_{MbaA}$ (Methods). We incorporated the function describing MbaA activity in a simple dynamical system considering polyamine fluxes into and out of the periplasm and the kinetics of MbaA-driven c-di-GMP biosynthesis and degradation. To acquire values for parameters, we fitted the model to the experimentally obtained c-di-GMP reporter data for the wild-type, Δ*nspS*, Δ*mbaA*, and Δ*potD1* strains (Fig 1B–1E, middle panels). Our rationale for choosing this particular model was that we could introduce experimentally titrated polyamine concentrations as inputs to the model, compare the modeled steady-state c-di-GMP concentrations to the experimentally obtained c-di-GMP reporter output data, and thus exploit the model to obtain the parameters required to achieve the measured input–output dynamics. To place additional constraints on the parameters, we experimentally measured the stoichiometry (approximately 1:1) of functional NspS-3xFLAG and MbaA-FLAG proteins in vivo (S1A and S1B Fig) as well as the apparent dissociation constants of spermidine

(48.4 ± 14.7 nM) and norspermidine (67.1 ± 12.2 nM) for purified NspS-6xHis using isothermal titration calorimetry (ITC) (S1C–S3E Fig).

Our model successfully captured the three regimes for the wild-type strain (Fig 1B, right panel), while achieving close fits to the data for the three mutants (Fig 1C–1E, right panels). The model fits suggest that in wild-type *V. cholerae*, the first regime is explained by two factors: (1) PotABCD1 imports norspermidine with high affinity ($K_m \sim 1$ nM); and (2) only a small fraction of NspS (approximately 0.7%) exists in the closed state in the absence of norspermidine (S1 Table). Supposition (1) is supported by experimental results from the Δ*potD1* strain, which show that in the absence of polyamine import, elevated periplasmic norspermidine drives maximal MbaA-directed c-di-GMP production (Fig 1E, middle panel). Supposition (2) is supported by the Δ*nspS* mutant experimental data, which show that in the absence of NspS, the basal c-di-GMP level is lower than that of the wild type (Fig 1C, middle panel). Thus, in the first regime, the model predicts that there is an inconsequentially low concentration of polyamines in the periplasm but that a small fraction of closed NspS nonetheless binds to MbaA and drives a modest level of c-di-GMP production (Fig 1B, right panel). To further test this prediction, we overexpressed *nspS* and *mbaA* from the native locus using the *Ptac* promoter, and we measured c-di-GMP reporter output across varying levels of norspermidine and spermidine. c-di-GMP reporter output was strikingly elevated compared to that in the wild-type strain, including in the absence of norspermidine. This result suggests that apo-NspS can bind MbaA and elicit diguanylate cyclase activity. In the second regime, our fitted kinetic parameters suggest that PotABCD1-mediated import of spermidine is saturated, resulting in the accumulation of periplasmic spermidine, which biases NspS to the open conformation (equivalent to an NspS-free state, as in Fig 1C). Therefore, this regime is explained by the intrinsic (NspS-free) fraction of MbaA receptors that exist in the phosphodiesterase mode, which our fits suggest is around 75% (S1 Table). In the third and final regime, our model suggests that the high concentration of exogenously supplied polyamines saturates PotABCD1-mediated import, and, therefore, MbaA activity is determined by a competition between periplasmic norspermidine and periplasmic spermidine binding to NspS. Using our ITC measurements (S1 Fig) and the predicted fraction of NspS that exists in the closed state (S1 Table), we calculated the dissociation constant $K_{nspd}$ for norspermidine for closed NspS to be approximately 0.5 nM and the dissociation constant $K_{spd}$ for spermidine for open NspS to be approximately 48.0 nM (see Methods for details concerning these calculations). The consequence of this approximately 100-fold difference in binding affinities is that once PotD1-mediated import is saturated by norspermidine, the additional periplasmic norspermidine biases a significant fraction of NspS toward the closed state, irrespective of the periplasmic concentration of spermidine over the range studied (Fig 1E, right panel). Thus, under this condition, MbaA acts as a diguanylate cyclase. Together, these results suggest that the NspS-MbaA circuit functions to maintain a basal cytoplasmic concentration of c-di-GMP via a mechanism in which a small fraction of total NspS binds MbaA in the absence of exogenous norspermidine. This arrangement may be necessary for the system to sensitively respond to both spermidine (and, in turn, degrade c-di-GMP) and norspermidine (and, in turn, synthesize c-di-GMP). In addition, high affinity uptake via PotABCD1 eliminates low-to-moderate concentrations of norspermidine and low concentrations of spermidine from the periplasm. The consequence is that under physiologically realistic regimes, the concentrations of polyamines in the periplasm are sub-nanomolar.

## The mathematical model for NspS-MbaA signal transduction is predictive

To test the fidelity of our model, we assessed its predictive capabilities by perturbing the NspS-MbaA signal transduction cascade. Specifically, we constructed a *V. cholerae* Δ*mbaA*

strain harboring *Pbad-mbaA*, which allowed us to artificially alter *mbaA* expression with arabinose. We modulated MbaA production from nondetectable (0% arabinose) to modestly below that of the wild type (0.05% arabinose), to higher than wild type levels (0.2% arabinose), as quantified by western blot (S2 Fig). We assessed how changing the level of MbaA influenced the steady-state c-di-GMP concentration by measuring the c-di-GMP reporter output for each MbaA induction condition and each polyamine concentration (Fig 2, left panels). We used our model to predict the c-di-GMP outputs under the same conditions (Fig 2, right panels). As expected, in the absence of arabinose, the experimental results and model predictions agreed (Fig 2A) and, moreover, produced results akin to those for the ΔmbaA strain (Fig 1D). Furthermore, our model predictions achieved close fits to the experimental c-di-GMP reporter outputs for both MbaA underexpression (Fig 2B) and overexpression (Fig 2C). Both under- and overexpression of MbaA dampened the response to norspermidine relative to the wild type, suggesting that the NspS-MbaA circuit is maximally sensitive to changes in norspermidine levels near the endogenous wild-type MbaA concentration and that norspermidine sensitivity decreases as MbaA concentrations deviate from this level. Together, our results indicate that our mathematical framework for the NspS-MbaA circuit accurately predicts the effects stemming from perturbations to receptor levels and that the natural system has evolved to be maximally sensitive to norspermidine when there is import by PotABCD1.

## The NspS-MbaA circuit is sensitive to sub-nanomolar periplasmic polyamine levels

Our data show that PotABCD1-mediated polyamine import dramatically influences the ability of NspS-MbaA to detect and respond to polyamines in the periplasm (Fig 1D). The consequence of rapid polyamine internalization is that polyamines (i.e., those supplied in experiments, those supplied by export from the cytoplasm, or those supplied by neighboring organisms in non-lab environments) are depleted from the periplasm. We reasoned that to overcome ligand depletion by cytoplasmic import, the NspS-MbaA circuit must be exquisitely sensitive to periplasmic polyamines. To test this supposition, we needed to quantify the function of NspS-MbaA in a setup in which norspermidine production and norspermidine and spermidine import were inactivated to eliminate cell-driven changes to the fixed levels of polyamines we supplied exogenously. Our strategy was to disable norspermidine production by deleting *nspC*, the gene encoding the carboxynorspermidine decarboxylase responsible for the final enzymatic step in norspermidine production [25]. To eliminate both polyamine production and import, we constructed the Δ*nspC* Δ*potD1* double mutant, a strain that has previously been used to assess the effect of exogenously supplied norspermidine on biofilm formation [22]. Analysis of these two mutant phenotypes allowed us to assess the intrinsic sensitivity of the NspS-MbaA circuit to periplasmic polyamines.

Regarding the Δ*nspC* single mutant that is incapable of norspermidine production: The model predicts that the *V. cholerae* Δ*nspC* mutant responses to external norspermidine and spermidine resemble those of the wild type (Fig 3A, right panel). This outcome occurs because, as discussed above, self-produced norspermidine exerts a negligible effect on basal MbaA activity in the presence of polyamine import (Fig 1A and 1B). Indeed, the middle panel of Fig 3A shows that the Δ*nspC* mutant exhibits the three c-di-GMP regimes displayed by the wild type (see Fig 1A), with an $EC_{50}$ for norspermidine of approximately 10 to 50 μM.

Regarding the Δ*nspC* Δ*potD1* double mutant that is incapable of norspermidine production and norspermidine and spermidine import: Our experimental results show that maximal c-di-GMP production was achieved at the lowest tested norspermidine concentration of 1 μM, and minimal c-di-GMP production occurred at 1 μM spermidine treatment (Fig 3B, middle

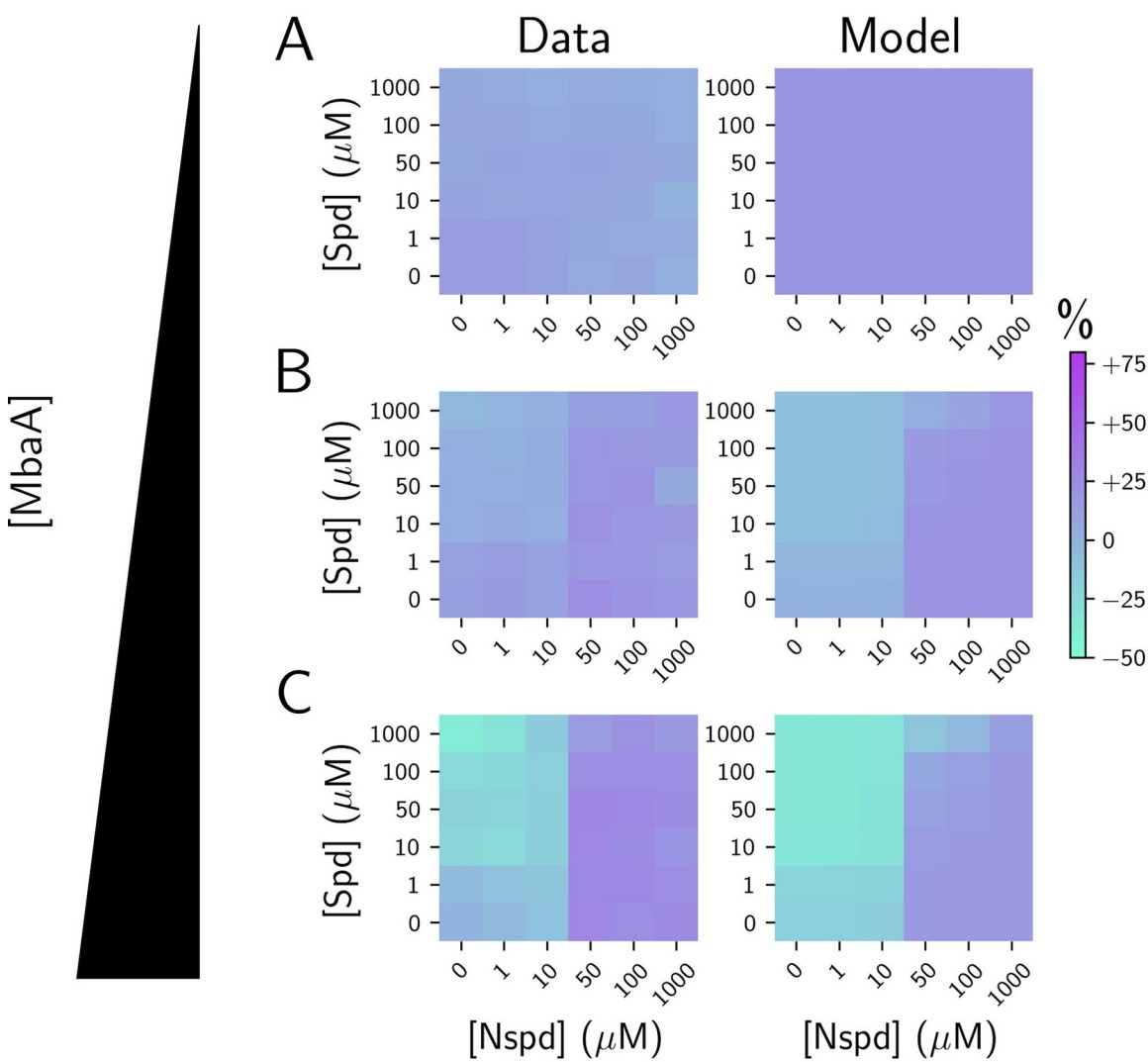

**Fig 2. The fitted mathematical model of NspS-MbaA input–output dynamics accurately predicts the outcomes of perturbations to MbaA protein levels.** (A) Left panel: experimentally obtained results for c-di-GMP reporter output in Δ*mbaA V. cholerae* carrying *Pbad-mbaA* for the indicated polyamine concentrations at 0% arabinose, shown as a heatmap. Throughout the manuscript, data in c-di-GMP output heatmaps are displayed as percent differences compared to the untreated wild-type strain, with teal representing low and purple representing high c-di-GMP reporter output, respectively. Right panel: modeled c-di-GMP reporter output for Δ*mbaA V. cholerae* based on a global fit to the experimental data for the wild-type, Δ*nspS*, Δ*mbaA*, and Δ*potD1* strains. (B) Left panel: as in A, left panel, but with supplementation of 0.05% arabinose. Right panel: modeled c-di-GMP reporter output for *V. cholerae* with approximately 2.5-fold lower MbaA levels than in wild-type *V. cholerae*. (C) Left panel: as in A, left panel, but with supplementation of 0.2% arabinose. Right panel: as in A, right panel, but for approximately 1.5-fold higher MbaA concentration than in wild-type *V. cholerae*. N = 3 biological replicates. Numerical values for plots are available in S1 Data. c-di-GMP, cyclic diguanylate; Nspd, norspermidine; Spd, spermidine.

panel). The consequence of sensitization to external polyamines is that the high c-di-GMP regime, in which norspermidine outcompetes spermidine, expands to include significantly lower norspermidine concentrations (Fig 3B, middle panel). The model output agreed with the experimental results, showing that the Δ*nspC* Δ*potD1* strain exhibits high sensitivity to exogenous spermidine and norspermidine relative to the Δ*nspC* strain due to the elimination of periplasmic polyamine depletion by PotABCD1 (Fig 3B, right panel). To define the intrinsic input–output relationship between periplasmic norspermidine and c-di-GMP levels, we

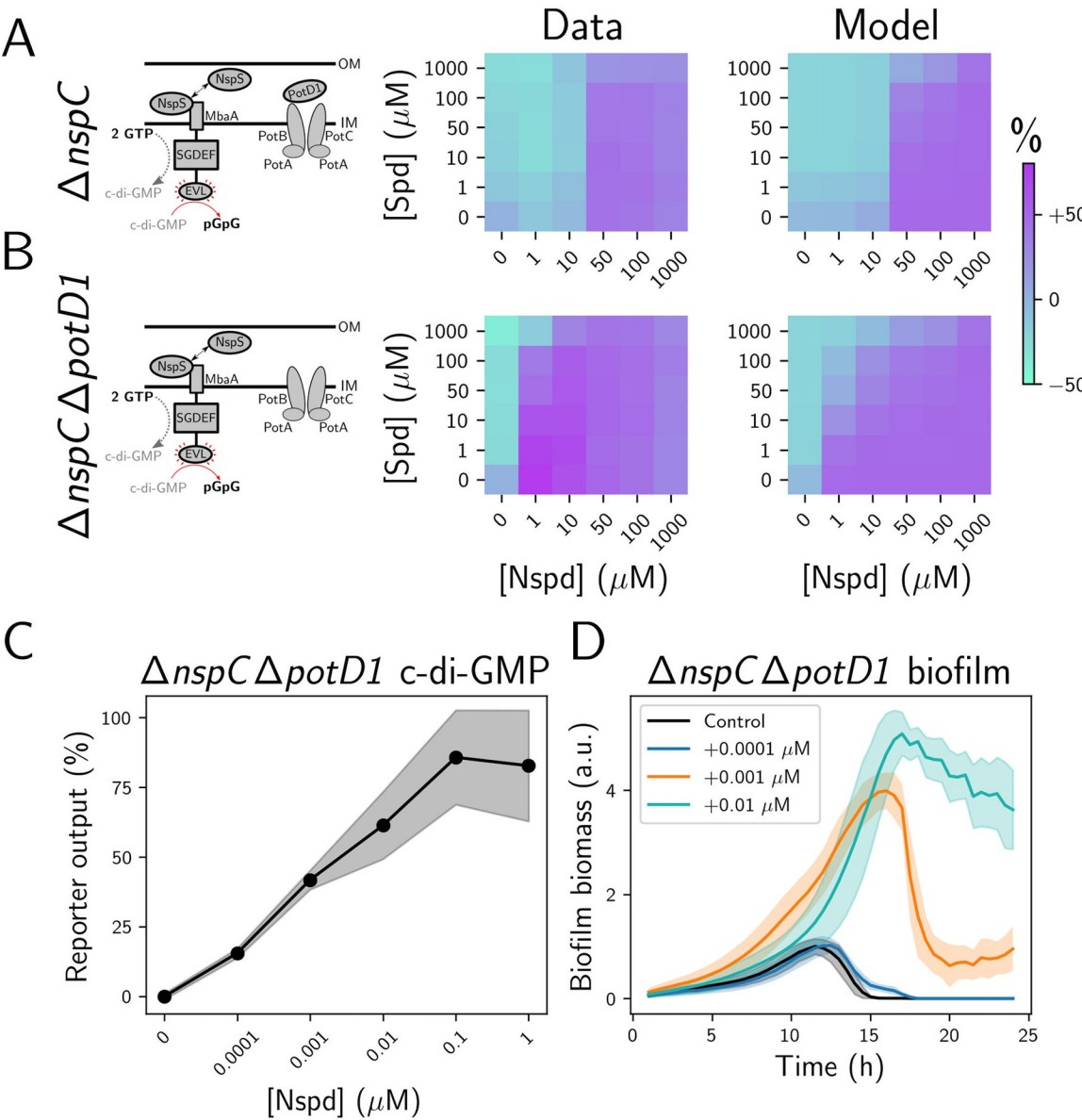

**Fig 3. MbaA-driven c-di-GMP production and biofilm formation are highly sensitive to Nspd in the absence of PotD1-mediated import.** (A) Left panel: schematic of the periplasmic polyamine sensing and import components in Δ*nspC V. cholerae* in the absence of exogenous polyamines. Middle panel: experimentally obtained results for the c-di-GMP reporter output, displayed as a heatmap, in Δ*nspC V. cholerae* for the indicated polyamine concentrations. *N* = 3 biological replicates. Right panel: modeled c-di-GMP reporter output for the Δ*nspC V. cholerae* strain. The parameters fitted to the data from Fig 1 were used. (B) As in A for Δ*nspC ΔpotD1 V. cholerae*. Throughout the manuscript, data in c-di-GMP output heatmaps are displayed as percent differences compared to the untreated wild-type strain, with teal representing low and purple representing high c-di-GMP reporter output, respectively. (C) Mean c-di-GMP output for Δ*nspC ΔpotD1 V. cholerae* at the specified Nspd concentrations. *N* = 3 biological replicates. (D) Biofilm biomass over time measured by bright field time-lapse microscopy for Δ*nspC ΔpotD1 V. cholerae* at the specified Nspd concentrations. *N* = 3 biological and *N* = 3 technical replicates, ± SD (shaded). Numerical values for plots are available in S1 Data. a.u., arbitrary unit; c-di-GMP, cyclic diguanylate; IM, inner membrane; Nspd, norspermidine; OM, outer membrane; Spd, spermidine.

supplied norspermidine to the Δ*nspC ΔpotD1* strain at concentrations below 1 μM and measured the reporter output. Fig 3C shows that a response occurred at a norspermidine concentration as low as 0.1 nM, and the response was saturated by 100 nM. Consistent with these results, we calculated an EC$_{50}$ for norspermidine of approximately 1 to 5 nM (Fig 3C). Thus, in

the absence of norspermidine import and export, the NspS-MbaA circuit is sensitive to sub-nanomolar changes in periplasmic norspermidine concentrations. Finally, the close agreement between the experimental results and the model for the Δ*nspC* single and the Δ*nspC* Δ*potD1* double mutants demonstrates that our model accurately captures the role of polyamine import in the response of the NspS-MbaA circuit to external norspermidine and spermidine.

Our model and experimental results indicate that the NspS-MbaA circuit is exceptionally sensitive to changes in periplasmic norspermidine levels. To investigate whether such sensitivity plays out via changes in downstream c-di-GMP-regulated behaviors, we assessed whether the *V. cholerae* Δ*nspC* Δ*potD1* mutant also exhibited altered biofilm formation and dispersal in response to sub-nanomolar concentrations of norspermidine. To do this, we administered 0.1 nM, 1 nM, and 10 nM norspermidine to the Δ*nspC* Δ*potD1* strain and assayed biofilm formation by time-lapse microscopy. Indeed, a dose-dependent increase in biofilm biomass occurred in response to norspermidine (Fig 3D). Together, these results show that sub-nanomolar levels of periplasmic norspermidine are detected by NspS-MbaA. Norspermidine detection leads to increased MbaA diguanylate cyclase activity and cytoplasmic c-di-GMP accumulation, the result of which is increased biofilm biomass accretion and reduced biofilm dispersal. Finally, the consequence of PotABCD1-directed polyamine internalization in wild-type *V. cholerae* is that cytoplasmic c-di-GMP levels are only altered in response to high concentrations of extracellular norspermidine (>10 μM; Fig 1B, middle panel), despite the remarkable sensitivity of the NspS-MbaA system to low concentrations of periplasmic polyamines.

## MbaA transmits information internally to elicit gene expression changes at polyamine concentrations well below those required to change the total concentration of cytoplasmic c-di-GMP

NspS-MbaA detection of norspermidine and spermidine leads to induction and repression, respectively, of *vps* biofilm matrix genes [16,17]. However, the precise input–output relationship between polyamine sensing, c-di-GMP levels, and downstream gene expression is not known. To explore the relationship, we supplied exogenous norspermidine and spermidine to wild-type *V. cholerae* carrying a *vpsL-lux* reporter and measured the bioluminescence output. As shown above, in wild-type *V. cholerae*, exogenous norspermidine and spermidine had no notable effects on measured c-di-GMP levels at low concentrations (≤10 μM) (Fig 1B, reproduced in Fig 4A). By contrast, even at the lowest tested concentration (1 μM), both polyamines drove significant changes in *vpsL* expression (Fig 4B). These changes depended on MbaA possessing c-di-GMP biosynthesis capability, as norspermidine and spermidine did not elicit alterations in *vpsL* expression in an MbaA SGAAF mutant that is defective for c-di-GMP biosynthesis (S3 Fig). To determine if the changes in *vps* expression tracked with changes in the biofilm lifecycle, we measured the biofilm biomass over time. Consistent with the *vpsL-lux* data, biofilm biomass increased or decreased following the addition of 1 μM norspermidine or spermidine, respectively (Fig 4C and 4D). Moreover, biofilm dispersal was inhibited at 10 μM exogenous norspermidine (Fig 4D). As noted, changes in global cytoplasmic c-di-GMP are only elicited when 50 μM or higher polyamines are supplied (Fig 4A and 4E). Thus, the dichotomy is that *vps* gene expression and the resulting changes in the *V. cholerae* biofilm lifestyle are triggered at extracellular norspermidine and spermidine levels that have no measurable effects on global cytoplasmic c-di-GMP levels, consistent with previous results [18]. These results suggest that at low polyamine concentrations, the NspS-MbaA circuit transmits c-di-GMP directly to transcription factors via a local mechanism, the outcome of which is modulation of *vps* expression without changes to global cytoplasmic c-di-GMP levels. By contrast, at high polyamine concentrations, NspS-MbaA–mediated signaling alters both *vps* expression

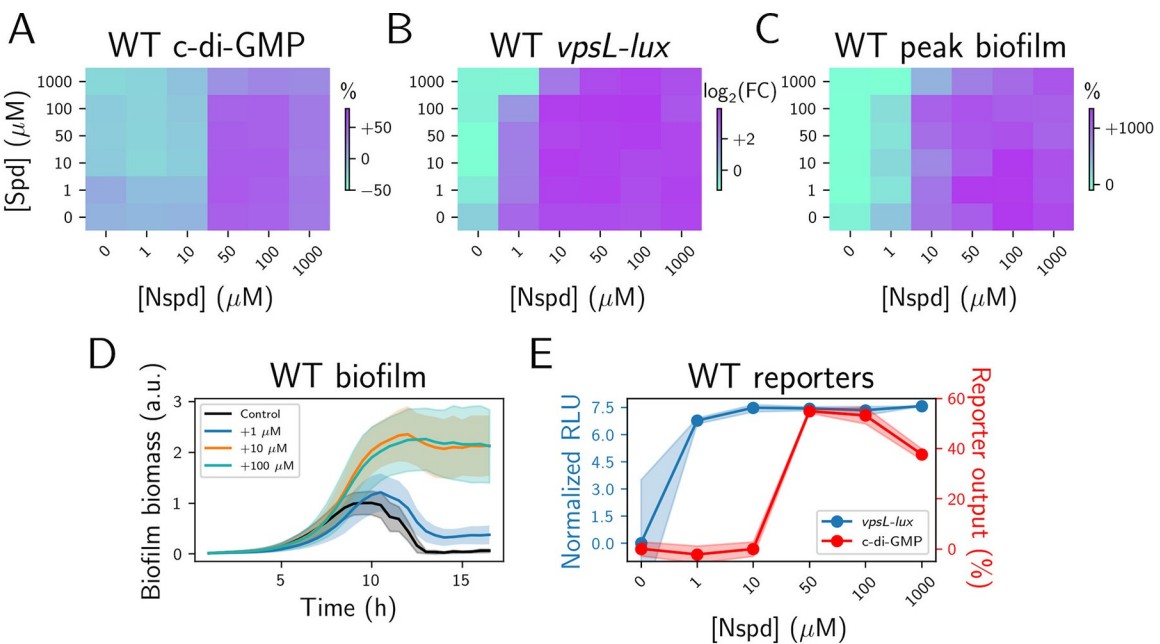

**Fig 4. *vps* gene expression and biofilm formation are more sensitive to external polyamines than is the cytoplasmic c-di-GMP pool.**
(A) Experimentally obtained results for c-di-GMP reporter output in wild-type *V. cholerae* for the indicated polyamine concentrations, displayed as a heatmap and reproduced from Fig 1A, middle panel. $N = 3$ biological replicates. (B) As in (A) for *vpsL-lux*. Values are displayed as the $\log_2$ fold changes relative to the untreated condition. (C) As in (B) for peak biofilm biomass measured by bright field time-lapse microscopy. $N = 2$ biological and $N = 2$ technical replicates. (D) Biofilm biomass over time measured by bright field time-lapse microscopy for wild-type *V. cholerae* at the specified Nspd concentrations $N = 3$ biological and $N = 3$ technical replicates, ± SD (shaded). (E) Mean *vpsL-lux* ($\log_2$ fold change relative to the untreated condition) and c-di-GMP outputs for wild-type *V. cholerae* at the specified Nspd concentrations. These plots were generated from the data points in the bottom rows of panels (A) and (B). Numerical values for plots are available in S1 Data. a.u., arbitrary unit; c-di-GMP, cyclic diguanylate; FC, fold change; Nspd, norspermidine; Spd, spermidine; *vps*, vibrio polysaccharide biosynthesis genes; WT, wild-type.

and the global cytoplasmic c-di-GMP reservoir, which could potentially be accessed by all c-di-GMP effectors.

## NspS-MbaA signal transduction specifically controls expression of biofilm genes, not genes involved in other c-di-GMP–regulated processes

Given that at high levels of norspermidine or spermidine, NspS-MbaA–directed signal transduction changes the cytoplasmic level of c-di-GMP, a diffusible second messenger known to regulate a wide variety of phenotypes, we wondered whether MbaA exclusively controls transcription of biofilm genes or more generally affects the expression of genes involved in other c-di-GMP–regulated processes, such as motility. To explore this question, we defined the NspS-MbaA–controlled regulon by comparing the transcriptome of untreated wild-type *V. cholerae* to that of wild-type *V. cholerae* treated with 100 μM exogenous norspermidine (Fig 5A and 5C) or 100 μM spermidine (Fig 5B and 5C). The data reveal that MbaA signaling is highly specific for control of biofilm gene transcription; following norspermidine treatment, 20 of the 31 significantly up-regulated genes ($\log_2$ fold change > 1 and $P$ value < 0.05 relative to untreated wild-type *V. cholerae*) encode *V. cholerae* biofilm matrix genes, operons, or known biofilm regulators (e.g., *vps-I*, *vps-II*, *rbm* cluster, and *vpsT*) (Fig 5A and 5C, S2 Data). Consistent with this finding, spermidine treatment caused down-regulation of these same genes (20 of the 25 down-regulated genes) (Fig 5B and 5C, S3 Data). Expression of a few other genes was altered in response to the polyamines, including *nspC*, genes encoding some

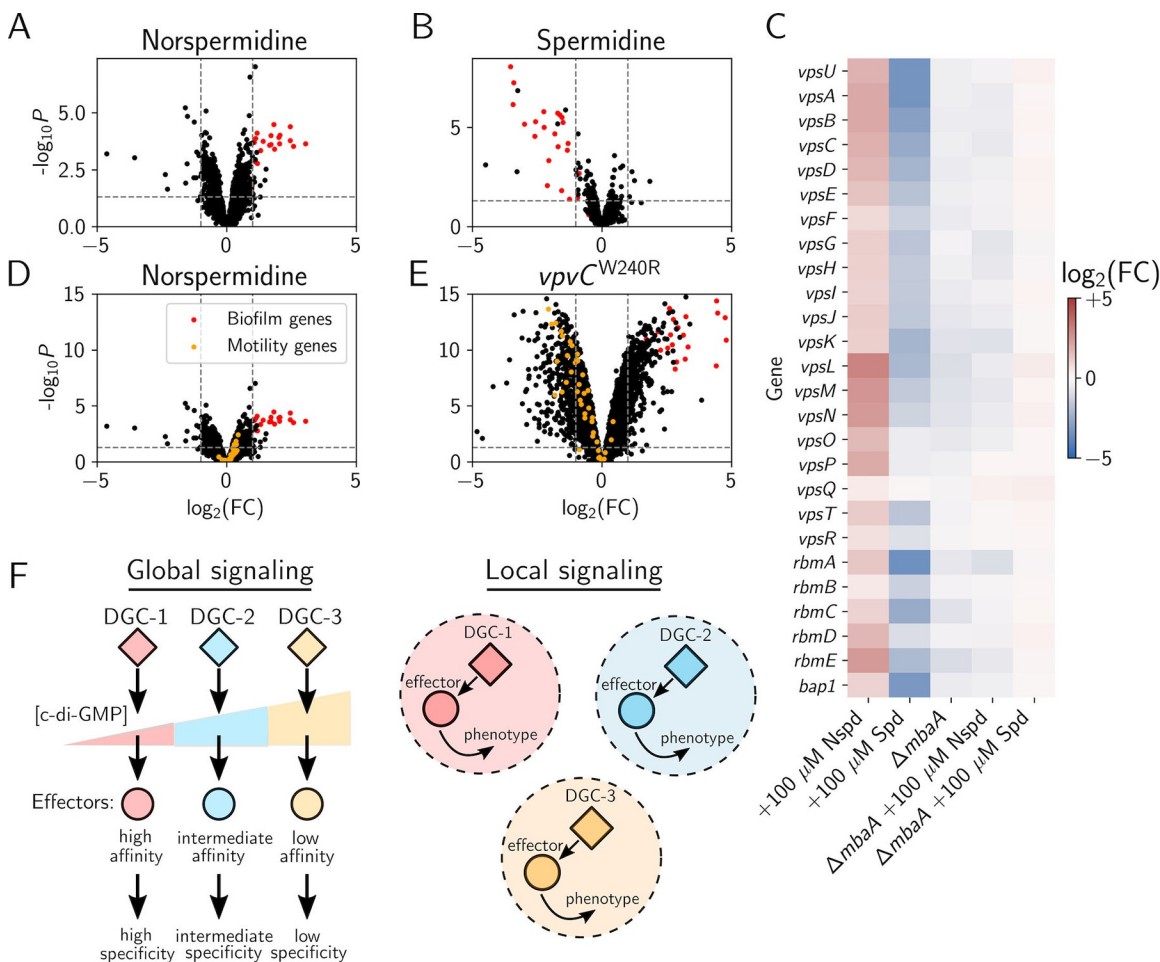

**Fig 5. Polyamine signaling through the NspS-MbaA circuit specifically controls biofilm gene expression.** (A) Volcano plot showing FCs in gene expression measured by RNA sequencing of the transcriptome of wild-type *V. cholerae* following administration of 100 μM Nspd relative to that of the untreated control. *vps* genes are highlighted in red, the horizontal dotted line represents a *p*-value of 0.05, and left and right vertical dashed lines represent $\log_2$ FCs of −1 and 1, respectively. Samples were collected at $OD_{600} = 0.1$ and $N = 3$ biological replicates. Complete datasets are available in S2–S11 Data. (B) As in A for 100 μM Spd. (C) Heatmap showing FCs in *V. cholerae* biofilm gene expression for the indicated treatments and strains relative to untreated wild-type *V. cholerae* (lanes 1–3) or the untreated Δ*mbaA* strain (lanes 4–5). Red and blue represent increased and decreased gene expression levels, respectively. (D) Identical data as in panel A, rescaled for ease of comparison to the *V. cholerae* *vpvC*^W240R strain. *vps* genes are highlighted in red, and motility genes are depicted in orange. (E) As in A, for the *V. cholerae* *vpvC*^W240R strain, FCs are relative to the untreated wild-type strain. (F) Schematic comparing the proposed global (left panel) and local (right panel) models to achieve specificity in c-di-GMP signaling circuits. See text for details. (Reviewed in [12,14]). Numerical values for plots are available in S1 Data. c-di-GMP, cyclic diguanylate; DGC, diguanylate cyclase. FC, fold change; Nspd, norspermidine; Spd, spermidine; *vps*, vibrio polysaccharide biosynthesis genes.

ribosome components, and some transporters. Crucially, however, no genes involved in other known c-di-GMP–regulated processes were significantly altered by polyamine treatment. We verified these findings by performing the same analyses on the Δ*mbaA* mutant: Norspermidine and spermidine treatment did not alter biofilm gene expression (Fig 5C, S1–S6 Data). Collectively, these data demonstrate that the two polyamines control the *V. cholerae* biofilm lifecycle exclusively through NspS-MbaA. Moreover, NspS-MbaA exerts an effect on only a small subset of c-di-GMP–responsive genes, notably *vps* genes, consistent with previous results [17].

We wondered if the ability to drive changes in only a select subset of c-di-GMP–responsive outputs is unique to NspS-MbaA or if other *V. cholerae* c-di-GMP metabolizing enzymes

regulate particular downstream genes with analogous selectivity. To address this question, we focused on specificity stemming from c-di-GMP synthesis by diguanylate cyclases. An especially attractive feature of the NspS-MbaA system is that MbaA diguanylate cyclase activity can be ramped up via exogenous supply of the norspermidine ligand. We are not aware of another individual *V. cholerae* diguanylate cyclase that can be controlled with high specificity by administration of an identified ligand. To overcome this issue, we exploited a *V. cholerae* strain that exhibits constitutive VpvC diguanylate cyclase activity due to a point mutation (W240R) [26]. The mutation "locks" *V. cholerae* in biofilm-forming mode and dispersal does not occur. Thus, the *V. cholerae vpvC*$^{W240R}$ mutant, at least phenotypically, mimics wild-type *V. cholerae* that has been supplied with norspermidine. Unlike wild-type *V. cholerae* treated with norspermidine (Fig 5D, rescaled data from 5A), *V. cholerae* carrying *vpvC*$^{W240R}$ exhibited broad changes in gene expression (725 genes were differentially expressed relative to wild type) (Fig 5E and S7 Data). The transcriptomic changes included higher activation of biofilm gene expression than that following norspermidine treatment of wild type (red points), and, additionally, repression of genes involved in cell motility (orange points), and changes to hundreds of genes required for other processes (black points). Thus, the c-di-GMP synthesized by VpvC$^{W240R}$ causes a dramatic and global reprogramming of *V. cholerae* gene expression, whereas the c-di-GMP produced by MbaA exclusively regulates biofilm genes.

Two mechanisms have been proposed to account for specificity in the output of c-di-GMP–responsive genes [14,27]. First is the "global signaling model" (Fig 5F, left panel), in which the c-di-GMP produced by a given diguanylate cyclase freely diffuses throughout the cytoplasm. Specificity in target gene expression is achieved by differences in affinities of downstream effectors for c-di-GMP and/or differences in effector affinities for target promoters [14]. For example, a weak diguanylate cyclase, represented by DGC-1 (Fig 5F, left panel), produces a low level of c-di-GMP. Only effectors with the highest affinities for c-di-GMP detect this change and, in turn, alter genes or proteins in specific pathways. In our schematic, diguanylate cyclase DGC-2, which makes more c-di-GMP than DGC-1, activates the DGC-1 effectors and additional lower-affinity effectors. Consequently, DGC-2 elicits changes in the expression of a larger set of c-di-GMP–regulated genes and the behaviors they specify than does DGC-1. Finally, a strong diguanylate cyclase, represented by DGC-3, produces the highest level of c-di-GMP, activating all c-di-GMP–responsive effectors, which, in turn, drive large-scale changes in gene expression and behavior. In this model, phosphodiesterases operate similarly except, when stimulated, they reduce the global c-di-GMP pool. The second model, termed the "local signaling model," (Fig 5F, right panel), posits that diguanylate cyclases and phosphodiesterases convey information only to particular downstream effectors, either through direct protein–protein interactions or via other means to produce local c-di-GMP pools [14]. In this model, specificity is achieved by directly ferrying c-di-GMP from the diguanylate cyclase enzyme to the effector(s) or by local c-di-GMP degradation in the case of phosphodiesterases.

## A global mechanism for c-di-GMP signaling delivers transcriptional specificity, but local c-di-GMP signaling via NspS-MbaA mediates increased polyamine sensitivity

We considered whether local and/or global c-di-GMP signaling mechanisms could explain the differences in transcriptional output specificity that occur due to c-di-GMP produced by NspS-MbaA (highly specific) and that by VpvC$^{W240R}$ (nonspecific). An obvious possibility in the context of the global c-di-GMP signaling mechanism is that activation of MbaA by norspermidine treatment only drives low-level production of cytoplasmic c-di-GMP, whereas VpvC$^{W240R}$ generates higher c-di-GMP concentrations. Thus, c-di-GMP produced by MbaA

only engages high-affinity biofilm regulatory effectors, while c-di-GMP made by VpvC$^{W240R}$ activates many more effectors. To test this possibility, we used the c-di-GMP reporter to compare the levels of c-di-GMP produced by the two diguanylate cyclases under the same conditions used for our transcriptomics measurements. The bottom panel of Fig 6 shows the results. Companion transcriptomics results are displayed above those data in heatmaps, categorized by pathway. Indeed, the *V. cholerae vpvC*$^{W240R}$ strain produced approximately 2- to 3-fold more c-di-GMP than wild-type *V. cholerae* treated with norspermidine. Thus, it is possible that differences in cytoplasmic c-di-GMP concentrations underpin the observed differences in gene expression outputs for the two diguanylate cyclases.

In the context of our analyses, the global c-di-GMP signaling model makes two predictions: (1) If MbaA-produced c-di-GMP were increased to match that of VpvC$^{W240R}$, a broader set of gene expression changes would occur, and specificity for biofilm regulation would be lost. (2) Conversely, if VpvC$^{W240R}$ c-di-GMP output were reduced to match that of MbaA following norspermidine treatment, then VpvC$^{W240R}$ would specifically regulate biofilm genes. To test whether these predictions hold, we synthetically modulated the amount of c-di-GMP made by MbaA and VpvC$^{W240R}$, measured cytoplasmic c-di-GMP levels, and assessed the transcriptomes. Importantly, the *V. cholerae* strain carrying the *vpvC*$^{W240R}$ allele does not produce levels of c-di-GMP that saturate the reporter. We know this because we can elicit higher reporter output by additionally overexpressing *Pbad-vpvC*$^{W240R}$ in that strain (S5 Fig). To address the first prediction, we increased c-di-GMP production by MbaA to match that of VpvC$^{W240R}$ by overexpressing *nspS* and *mbaA* using the *Ptac* promoter (Fig 6). Consistent with the global model, *V. cholerae* overexpressing *nspS* and *mbaA* no longer exhibited transcriptional specificity. Rather, 624 genes were differentially regulated relative to wild type. Notably, biofilm genes showed strong activation and motility genes showed strong repression (Figs 6 and S4, S8 Data). To address the second prediction, we controlled the expression of *vpvC*$^{W240R}$ using the arabinose-controllable *Pbad* promoter (Figs 6 and S4, S9 Data). Overall, gene expression changed in step with the amount of VpvC$^{W240R}$ produced, as dictated by the arabinose inducer concentration. At the arabinose concentration at which the c-di-GMP output most closely matched that of wild-type *V. cholerae* treated with 100 μM norspermidine, the two strains demonstrated nearly identical transcriptomic profiles (Fig 6, boxed). Due to short induction times, the arabinose inducible *vpvC*$^{W240R}$ construct did not enable us to achieve the sustained high c-di-GMP concentrations needed for repression of motility genes. Nonetheless, our results suggest that the global c-di-GMP signaling model could explain the specificity of the NspS-MbaA pathway for biofilm gene regulation and the lack of specificity in the VpvC$^{W240R}$ strain.

The above findings suggest that any c-di-GMP diguanylate cyclase/phosphodiesterase enzyme could deliver transcriptional specificity if its levels were appropriately modulated. To determine if this is the case, we used our arabinose induction strategy to modulate the levels of CdgL, another *V. cholerae* diguanylate cyclase. We know that CdgL contributes to the basal c-di-GMP pool because the Δ*cdgL* mutant possesses reduced cytoplasmic c-di-GMP and exhibits lower biofilm gene expression than does wild-type *V. cholerae*. Indeed, the Δ*cdgL* mutant behavior is akin to that of wild-type *V. cholerae* treated with 100 μM spermidine (Figs 6 and S4, S10 Data). We discovered *cdgL* induction conditions in which the cytoplasmic c-di-GMP level equaled that of wild-type *V. cholerae* treated with 100 μM norspermidine (Fig 6, boxed). Again, the transcriptional profile of the strain with synthetically modulated CdgL levels mirrored that of wild type following norspermidine treatment (Figs 6 and S4, S10 Data). Together, these results show that c-di-GMP signaling specificity can be achieved exclusively through changes to the global cytoplasmic pool. Thus, local signaling is not required for specificity in transcription.

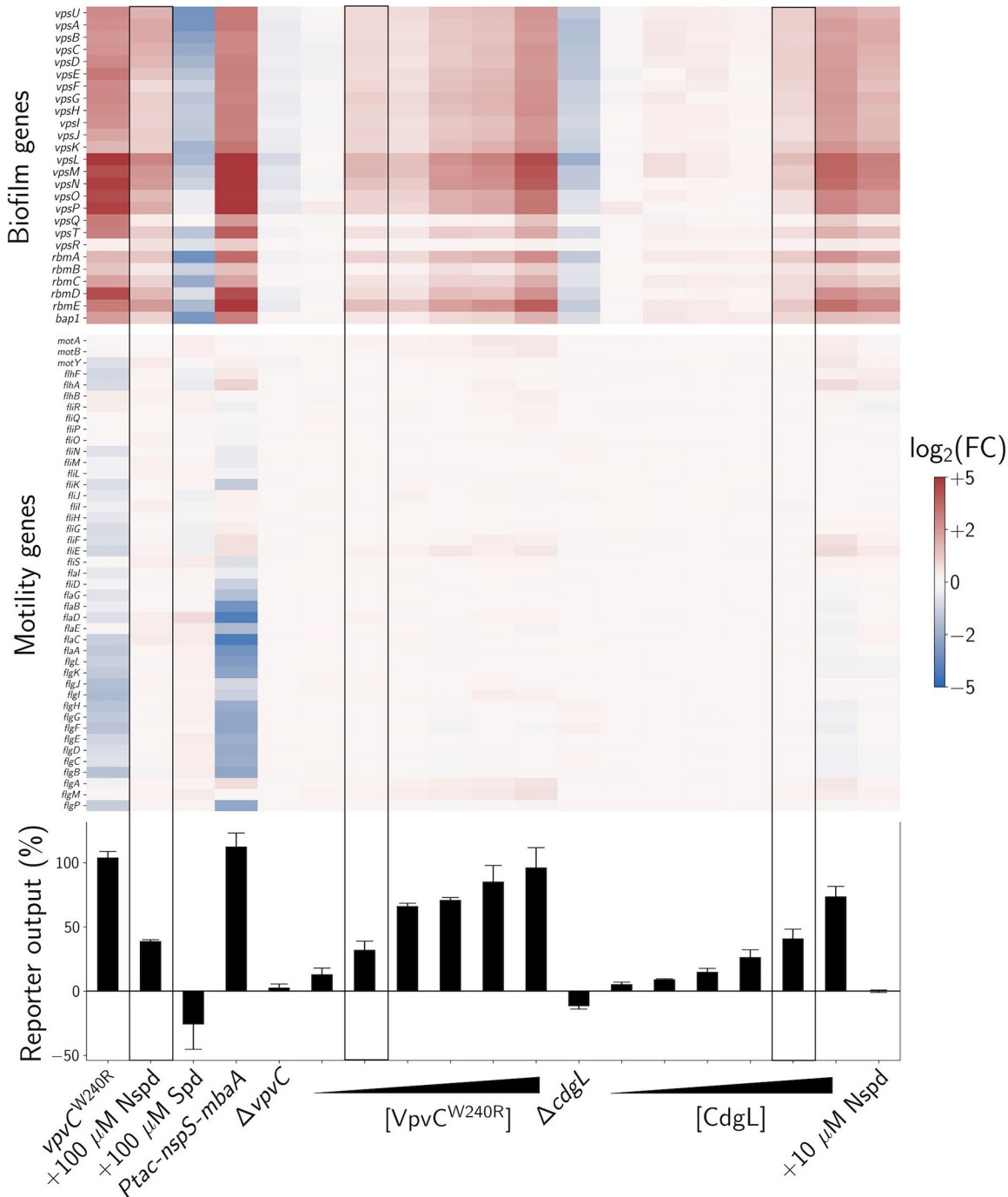

**Fig 6. Consequences of global and local c-di-GMP changes on *V. cholerae* gene expression patterns.** Bottom panel: mean global c-di-GMP reporter outputs for the indicated strains and conditions expressed as percentage differences relative to untreated wild-type *V. cholerae*. $N$ = 3 biological replicates. Strains carrying cloned genes encoding enzymes that synthesize c-di-GMP were grown with 0%, 0.0125%, 0.025%, 0.0375%, 0.05%, or 0.1% arabinose to increasingly induce expression. To prevent excessive biofilm formation from interfering with reporter measurements, we deleted the *vpsL* gene from the *vpvC*^*W240R*^ and *Ptac-nspS-mbaA* strains. Data were normalized to the Δ*vpsL* parent strain carrying the c-di-GMP reporter. Top panel: heatmap of log₂ FCs for *V. cholerae* biofilm gene expression for the conditions and strains shown in the bottom panel. Genes are grouped by function (biofilm and motility), and red and blue represent increased and decreased expression levels, respectively. Boxed regions designate the gene expression outputs at approximately equal cytoplasmic c-di-GMP levels following Nspd treatment, arabinose induction of *vpvC*^*W240R*^, and arabinose induction of *cdgL*. Samples were collected at $OD_{600}$ = 0.1, complete datasets are available in S1 Data, and volcano plots for each condition are shown in S4 Fig. For transcriptomics studies, the *vpvC*^*W240R*^ and *Ptac-nspS-mbaA* strains carried intact *vpsL*. Numerical values for plots are available in S1 Data. c-di-GMP, cyclic diguanylate; FC, fold change; Nspd, norspermidine; Spd, spermidine.

The consistency of our data with the global c-di-GMP signaling model suggests that the MbaA phosphodiesterase should be able to degrade c-di-GMP produced by a different enzyme, such as VpvC$^{W240R}$. To test this possibility, we administered arabinose to *V. cholerae* carrying *Pbad-vpvC$^{W240R}$* to achieve c-di-GMP reporter output equal to that of wild-type *V. cholerae* treated with 100 µM norspermidine, as described above and shown in Fig 6. We subsequently titrated in either norspermidine or spermidine and assessed whether, in response to the ligands, MbaA altered the cytoplasmic concentration of c-di-GMP (S6 Fig). Indeed, norspermidine treatment elevated the c-di-GMP level, while spermidine treatment reduced the c-di-GMP level to roughly that of untreated wild type levels (S6 Fig). These results further demonstrate the global nature of c-di-GMP signaling by showing that MbaA can contribute to and deplete the global cytoplasmic c-di-GMP pool, the level of which is set by the combined activities of the suite of *V. cholerae* diguanylate cyclases and phosphodiesterases. Collectively, our results suggest that in *V. cholerae*, specificity in the biofilm gene expression output response to c-di-GMP signaling does not require local c-di-GMP signaling and can be achieved via global changes in cytoplasmic c-di-GMP levels (Fig 6).

The existence of global c-di-GMP signaling does not eliminate the possibility that local c-di-GMP signaling could also take place. Indeed, our data indicate that MbaA transmits c-di-GMP directly to select downstream biofilm effectors through a local mechanism, as evidenced by activation of *vpsL-lux* (Fig 4) and other genes encoding components required for biofilm formation (Figs 6 and S4, S11 Data) at norspermidine concentrations that are too low (≤10 µM) to elicit increases to the global cytoplasmic c-di-GMP pool (Fig 4). The proposed local signaling mechanism employed by the NspS-MbaA circuit sensitizes *V. cholerae* to submicromolar concentrations of norspermidine. This feature of the NspS-MbaA system presumably allows *V. cholerae* to modify its biofilm lifecycle in response to environmentally encountered spermidine and norspermidine concentrations. Fig 7 presents a schematic that reconciles the findings presented here and offers a model for how ligand sensitivity and signaling specificity are achieved through both local and global signaling.

## Discussion

In this study, we performed a quantitative analysis of signal transmission via the second messenger molecule c-di-GMP by characterizing the NspS-MbaA polyamine signaling circuit in *V. cholerae*. Our work uncovers how the pathway functions, from ligand binding to behavioral output. The parameter values from our mathematical model, which are underpinned by experimental data, suggest that high-affinity import depletes nearly all norspermidine from the periplasm of wild-type *V. cholerae*. As a result, NspS is unliganded; however, the parameter values also suggest that a small fraction of apo-NspS exists in the conformation that can bind MbaA, resulting in slight MbaA diguanylate cyclase activity and consequently some production of cytoplasmic c-di-GMP. If apo-NspS could not bind MbaA, MbaA would exhibit maximal phosphodiesterase activity (akin to the experimental results from Fig 1C), and *V. cholerae* would not respond to fluxes of spermidine. On the other hand, if a large fraction of apo-NspS could bind to MbaA, MbaA would exhibit maximal diguanylate cyclase activity (akin to the experimental results from Fig 1E), and *V. cholerae* would not respond to fluxes in norspermidine. Thus, a key takeaway from our modeling is that confining apo-NspS-MbaA complex formation to a low level underlies the capacity of *V. cholerae* cells to respond to both norspermidine and spermidine. In the future, the predictive capability of our model could be used, most powerfully in conjunction with genetic or synthetic biology approaches, to further characterize polyamine-mediated control of *V. cholerae* biofilms.

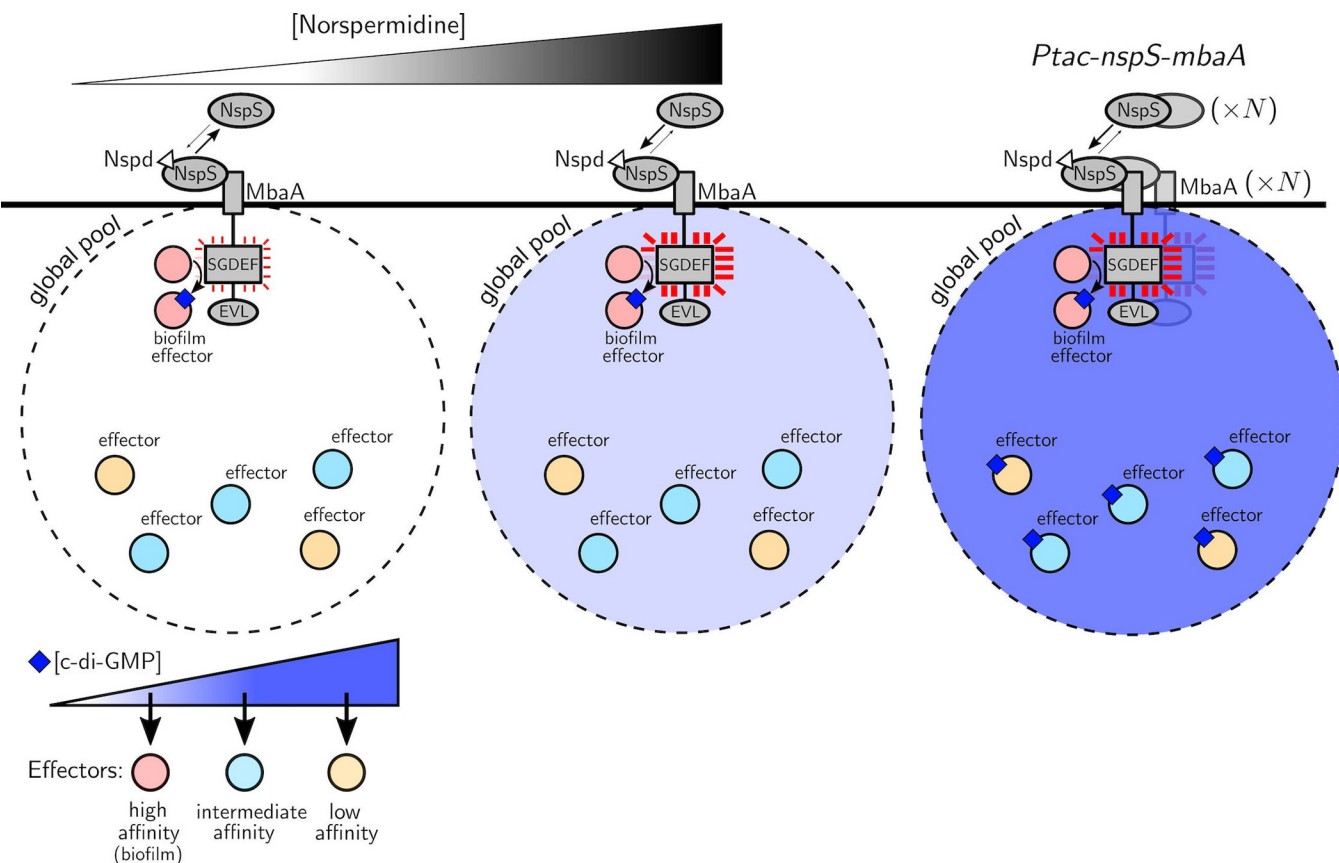

**Fig 7. Model for local and global c-di-GMP signaling for the NspS-MbaA pathway in *V. cholerae*.** At low norspermidine concentrations, NspS-MbaA transmits c-di-GMP directly to particular high affinity biofilm effectors via a local mechanism. This local mechanism sensitizes *V. cholerae* to norspermidine. At high norspermidine concentrations, when the NspS-MbaA diguanylate cyclase activity is maximal, the level of c-di-GMP that MbaA produces surpasses the amount that can be accommodated in the local pool. The extra c-di-GMP is contributed to the global pool and in principle becomes accessible to additional effectors. However, as shown in the results in Fig 6, only the subset of transcription effectors with the highest affinity for c-di-GMP, i.e., those regulating biofilm gene expression, detect these low-level changes to the global c-di-GMP pool. In the case of NspS-MbaA signaling, it appears that this subset of effectors is saturated by the local signaling mechanism. Thus, specificity in biofilm gene expression output is retained across all norspermidine ligand levels (Fig 6). Under artificial conditions, such as overexpression of *nspS* and *mbaA* by *Ptac*, the global c-di-GMP pool is driven higher, lower affinity effectors that control genes other than those involved in biofilms are engaged, and gene expression output specificity is lost.

Bacterial species frequently possess dozens of receptors harboring c-di-GMP biosynthetic and catabolic activities, underscoring the widespread and conserved nature of these signaling pathways [13]. Because of the central importance of c-di-GMP–based regulation of bacterial lifestyle decision-making processes, c-di-GMP pathways have been proposed as potential targets for the development of therapeutics that modify bacterial behavior [6]. Success in such an endeavor could be accelerated by quantitative understanding of the input–output relationships for specific c-di-GMP signaling circuits. A significant obstacle to progress is that for the vast majority of c-di-GMP signaling systems, the ligands that stimulate the receptors remain unknown [7]. Nevertheless, for c-di-GMP metabolizing receptors with known ligands, such as VC1086 and CdpA (nitric oxide), VC1710 (sugars), and CdgH (arginine) in *V. cholerae*, combining mathematical modeling with measurements of changes in c-di-GMP levels and downstream transcriptional reporter assays could be immediately undertaken and possibly reveal the interactions driving the observed input–output relations [28–31].

In addition to the modeling effort, we used the NspS-MbaA pathway as a case study to explore the roles of local and global c-di-GMP signaling pools. We find that c-di-GMP–driven specificity in transcription of biofilm genes in *V. cholerae* does not require localized signaling, as evidenced by our results following titrations of VpvC$^{W240R}$ and CdgL (Fig 6). This result is presumably a consequence of effectors with the highest affinity for c-di-GMP accessing the global pool and, in turn activating transcription of biofilm genes (Figs 4, 6, and 7). Nonetheless, our results strongly imply that NspS-MbaA must also be capable of transmitting c-di-GMP directly to particular effectors to alter downstream *vps* expression at low extracellular polyamine ligand levels that are insufficient to affect the concentration of the global c-di-GMP pool (Fig 7). Because the output gene expression pattern is identical under conditions in which NspS-MbaA signals exclusively locally (i.e., 10 μM norspermidine) or locally and globally (i.e., when NspS-MbaA is saturated at 100 μM norspermidine), we presume that MbaA signals locally to the same set of effectors (i.e., those responsible for regulating biofilm genes) that detect changes to the global c-di-GMP levels with high affinity. Thus, in the case of the NspS-MbaA system, while localized signaling is not required for specificity, it does confer higher sensitivity to polyamines than could be achieved if the relevant downstream effectors only responded to MbaA-driven changes to the global c-di-GMP pool (Figs 4 and 7). It is possible that in other c-di-GMP signaling systems, the effectors responsible for conveying local signals differ from the high-affinity global effectors. In such cases, output specificity, or lack thereof, could be governed by stimuli levels: At low stimulus concentrations, specificity could be achieved through direct c-di-GMP transmission to a local partner effector, analogous to the scenario we present for NspS-MbaA. However, at high stimulus concentrations, the c-di-GMP produced could exceed the amount required to saturate the local effectors, that c-di-GMP would leak into the global c-di-GMP pool, alter its levels, and drive broad changes to gene expression. Thus, unlike for NspS-MbaA, output specificity would decline with increasing stimulus concentration.

To prove the existence of the NspS-MbaA localized mechanism, it will be necessary to define the downstream effectors that partner with NspS-MbaA. While not verified at present, we suspect that the c-di-GMP responsive transcription factors VpsT and/or VpsR are the key effectors and they are the focus of our ongoing work in this direction [32,33]. VpsT and VpsR are the master regulators of *V. cholerae* biofilm gene expression and could interact with the MbaA catalytic domains [34]. In this regard, VpsT could act doubly as a c-di-GMP effector as it is also known to regulate motility [33]. Direct interactions between c-di-GMP metabolizing enzymes and downstream effectors have been demonstrated in other systems [12]. It is also worth noting that in the current work, we restricted our analyses to the transcriptional output in response to NspS-MbaA–directed c-di-GMP signal transduction. It remains possible that NspS-MbaA also controls c-di-GMP–dependent processes by posttranscriptional mechanisms, again acting either locally or globally.

It is increasingly appreciated that from bacteria to humans, second messenger molecules have the remarkable capacity to signal with high specificity despite their general use in an array of processes in the same cell and in the face of their high diffusivity. Indeed, as one example, calcium signaling in eukaryotes relies on many of the principles germane to c-di-GMP signaling in bacteria: Calcium signal transduction lies at the core of cell physiology and function, there exist multiple sources of the calcium messenger molecule, it is diffusible, and a large set of effectors respond to changes in its levels [35]. A rich body of literature demonstrates that calcium signaling fidelity is achieved via formation of local microdomains that promote directed signal transmission [36]. Thus, in different guises, evolution has solved the same issues associated with generically used, diffusible second messenger signaling by devising mechanisms for locally restricting signal transduction.

## Methods

### Model description

To describe the relation between periplasmic polyamine concentrations and c-di-GMP output from the NspS-MbaA circuit, we developed a two-state receptor model for MbaA. We assume that MbaA exists in one of two states: the diguanylate cyclase state or the phosphodiesterase state, which presumably correspond to distinct conformations of the MbaA homodimer. The average phosphodiesterase activity of MbaA, $\langle A_{\text{PDE}} \rangle$, is equivalent to the probability of being in the phosphodiesterase state, which in the equilibrium statistical mechanical description is determined exclusively by the free-energy offset between the phosphodiesterase and diguanylate cyclase states, $f_{\text{MbaA}}$ (with all energies in units of the thermal energy $k_{\text{B}}T$):

$$\langle A_{\text{PDE}} \rangle = \frac{1}{1 + \exp(f_{\text{MbaA}})} \tag{1}$$

As a consequence, the diguanylate cyclase activity of MbaA, $\langle A_{\text{DGC}} \rangle$, is given by $\langle A_{\text{DGC}} \rangle = 1 - \langle A_{\text{PDE}} \rangle$. $f_{\text{MbaA}}$ is a function of the intrinsic (NspS-free) free-energy offset between the phosphodiesterase and diguanylate cyclase states, $\epsilon_{\text{MbaA}}$, the concentration of free NspS in the closed conformation, $N_{\text{free}}^{\text{closed}}$, and the binding constant of MbaA in the diguanylate cyclase state for the closed conformation of NspS, $K_{\text{MbaA}}$:

$$f_{\text{MbaA}} = \epsilon_{\text{MbaA}} + \log\left[1 + \frac{N_{\text{free}}^{\text{closed}}}{K_{\text{MbaA}}}\right]. \tag{2}$$

$N_{\text{free}}^{\text{closed}}$, in turn, is a function of $\epsilon_{\text{MbaA}}$ and $K_{\text{MbaA}}$, as well as the intrinsic free-energy offset between the open and closed states of apo-NspS ($\epsilon_{\text{NspS}}$), periplasmic norspermidine and spermidine concentrations ($n_{\text{peri}}$ and $s_{\text{peri}}$, respectively), the molar ratio of MbaA to NspS ($R$), the binding constants of each NspS conformation for norspermidine and spermidine ($K_{\text{nspd}}$ and $K_{\text{spd}}$, respectively), and the total concentration of NspS ($N$) (see S1 Text for the full derivation):

$$N_{\text{free}}^{\text{closed}} = N \frac{e^{\epsilon_{\text{NspS}}} \left(\frac{1 + n_{\text{peri}}/K_{\text{nspd}}}{1 + s_{\text{peri}}/K_{\text{spd}}}\right)}{1 + e^{\epsilon_{\text{NspS}}} \left(\frac{1 + n_{\text{peri}}/K_{\text{nspd}}}{1 + s_{\text{peri}}/K_{\text{spd}}}\right)} \left(1 - \frac{R e^{\epsilon_{\text{MbaA}}} \frac{N_{\text{free}}^{\text{closed}}}{K_{\text{MbaA}}}}{e^{\epsilon_{\text{MbaA}}} \left(\frac{N_{\text{free}}^{\text{closed}}}{K_{\text{MbaA}}} + 1\right) + 1}\right). \tag{3}$$

We incorporated $\langle A_{\text{PDE}} \rangle$ and $\langle A_{\text{DGC}} \rangle$ in a system of ordinary differential equations, which model the effect of a constant extracellular polyamine source on the steady-state proportions of MbaA homodimers in the phosphodiesterase and diguanylate cyclase states and the effect of MbaA activity on the cytoplasmic c-di-GMP concentration (for parameter definitions, see S1 Table):

$$\frac{dn_{\text{peri}}}{dt} = \alpha + \beta_n(n_{\text{ext}} - n_{\text{peri}}) - \frac{\psi P n_{\text{peri}}/K_{\text{PotD1}}^n}{1 + \frac{s_{\text{peri}}}{K_{\text{PotD1}}^s} + \frac{n_{\text{peri}}}{K_{\text{PotD1}}^n}} \tag{4}$$

$$\frac{ds_{\text{peri}}}{dt} = \beta_s(s_{\text{ext}} - s_{\text{peri}}) - \frac{\varphi P s_{\text{peri}}/K_{\text{PotD1}}^s}{1 + \frac{s_{\text{peri}}}{K_{\text{PotD1}}^s} + \frac{n_{\text{peri}}}{K_{\text{PotD1}}^n}} \tag{5}$$

$$\frac{dc}{dt} = \gamma + \lambda\langle A_{\text{DGC}} \rangle - (\nu + \mu\langle A_{\text{PDE}} \rangle)c. \tag{6}$$

## Model fitting procedure

To fit the mathematical model to our experimentally obtained c-di-GMP reporter assay data, we set $n_{\text{ext}}$ and $s_{\text{ext}}$ equal to the experimentally supplied concentrations of norspermidine and spermidine, respectively, initialized the state variables, and simulated (3–5) to a steady state over a range of parameter values. To optimize the parameter values, we used nonlinear least squares, which seeks a minimizing vector of parameter values, $x^*$, for a nonlinear objective function, $F$, of the form

$$\min_{x} F(x) = \min_{x} \frac{1}{2} ||r(x)||_2^2 = \min_{x} \frac{1}{2} \sum_{i=1}^{m} r_i(x)^2, \tag{7}$$

where $r_i(x)$ are the residuals representing the offset between each measured data point ($y_i$) and the modeled steady-state c-di-GMP output ($c^*$) when the supplied norspermidine and spermidine concentrations are $n_i$ and $s_i$, respectively:

$$r_i(x) = y_i - qc^*(x, n_i, s_i). \tag{8}$$

In (8), the model c-di-GMP output is scaled by a constant $q$, because the c-di-GMP reporter responds linearly with cytoplasmic c-di-GMP concentration [23]. To perform this optimization task, we implemented the Levenberg–Marquardt algorithm through the lmfit package in Python 3 [37] (see S1 Text for further details). The fitted parameter values are shown in S1 Table.

## Calculating the $K_{\text{d}}$ values for the different NspS conformations from ITC measurements

We calculated the binding affinity of norspermidine for the closed conformation of NspS ($K_{\text{nspd}}$) based on the equilibrium fraction of NspS bound to norspermidine. In general, in the absence of spermidine, this fraction is given by

$$p_{\text{bound}} = \frac{e^{\epsilon_{\text{NspS}}} \frac{n_{\text{peri}}}{K_{\text{nspd}}}}{1 + e^{\epsilon_{\text{NspS}}} \left( \frac{n_{\text{peri}}}{K_{\text{nspd}}} + 1 \right)}, \tag{9}$$

where $\epsilon_{\text{NspS}}$ denotes the intrinsic free-energy offset between the open and closed conformations of NspS. Thus, the relation between the measured apparent binding affinity, $K_{\text{d}}^n$, and the closed conformational binding affinity, is given by

$$K_{\text{d}}^n = \frac{K_{\text{nspd}}(1 + \exp(\epsilon_{\text{NspS}}))}{\exp(\epsilon_{\text{NspS}})}. \tag{10}$$

Similarly, the relation between the measured binding affinity of spermidine for NspS, $K_{\text{d}}^s$, and the binding affinity of spermidine for open NspS, $K_{\text{spd}}$, is given by

$$K_{\text{d}}^s = K_{\text{spd}}(1 + \exp(\epsilon_{\text{NspS}})). \tag{11}$$

## Bacterial strains, reagents, reporters, imaging assays, and western blots

The wild-type *V. cholerae* parent used in this work was *V. cholerae* O1 El Tor biotype C6706str2. All strains used in this work are reported in S2 Table. When necessary, antimicrobials were supplied at the following concentrations: polymyxin B, 50 μg/mL; kanamycin, 50 μg/mL; spectinomycin, 200 μg/mL; chloramphenicol, 10 μg/mL; and gentamicin, 15 μg/mL. Strains used for cloning were propagated on lysogeny broth (LB) plates supplemented with

1.5% agar or in liquid LB with shaking at 30˚C. Strains used in reporter quantitation, biofilm assays, and RNA isolation were grown in M9 minimal medium with 0.5% dextrose and 0.5% casamino acids. Norspermidine (Millipore Sigma, I1006-100G-A), spermidine (Millipore Sigma, S2626-1G), and arabinose (Millipore Sigma, W325501) were added at the concentrations designated in the figures or figure legends at the start of each assay. c-di-GMP, *vpsL-lux*, and biofilm biomass over time were measured as previously described [16,38,39]. The heatmap of peak biofilm biomass in Fig 4 employed bright field images that were obtained using a Biotek Cytation 7 imaging plate reader and a 20x objective lens. All plots were generated using Python 3 [37]. Western blotting of MbaA-3xFLAG and NspS-3xFLAG was performed as described previously [16].

## DNA manipulation and strain construction

Modifications to the *V. cholerae* genome were generated by replacing genomic DNA with linear DNA introduced by natural transformation as described previously [39]. PCR and Sanger sequencing (Genewiz) were used to verify results. See S3 Table for primers and g-blocks (IDT) used in this study. Due to overlap between the *nspS* and *mbaA* coding regions, gene synthesis was used to simultaneously produce *nspS-3xFLAG* at the endogenous locus and preserve the downstream coding sequence of *mbaA*. To achieve this arrangement, the *nspS-mbaA* overlapping region was duplicated. The *nspS* codon usage was altered in the upstream duplication while preserving the amino acid sequence, and the 5′-most *mbaA* start codon was disabled by mutation. DNA encoding *3xFLAG* was introduced immediately upstream of the *nspS* stop codon. DNA specifying a flexible linker (5x glycine, serine repeats) was inserted between the DNA encoding the 3′ terminus of NspS and the 5′ start of the DNA encoding the 3xFLAG tag. We found that such a linker was required for function. Arabinose titratable *Pbad* expression constructs were introduced at the neutral locus, *vc1807*. Reporters expressed on plasmids were introduced into *V. cholerae* strains via conjugation with *E. coli* S17 λ*pir*.

## NspS protein purification and isothermal titration calorimetry

DNA encoding NspS-6xHis, excluding its secretion signal (residues 1 to 34), was cloned into the pET-15b vector using Gibson assembly (NEB). Production of NspS-6xHis protein was initiated in *E. coli* BL21 (DE3) by the addition of 1 mM IPTG, followed by growth with shaking for 20 h at 18˚C. The cells were pelleted at $16,000 \times g$ for 20 min and resuspended in lysis buffer (50 mM Tris–HCl (pH 8.0), 150 mM NaCl, 10 mM imidazole, 1 mM DTT, 0.2 mg/mL lysozyme, 25 u/mL benzonase nuclease, and 1x EDTA-free protease inhibitor cocktail (Roche)). The cells were lysed using sonication and subjected to centrifugation at $32,000 \times g$ for 30 min. NspS-6xHis protein was purified from the clarified supernatant using Ni-NTA Superflow resin (Qiagen) equilibrated in lysis buffer. The column was washed three times with 10x column volumes of wash buffer (50 mM Tris–HCl (pH 8.0), 300 mM NaCl, 20 mM Imidazole, 1 mM DTT, and 1x EDTA-free protease inhibitor cocktail) and the protein was eluted with the same buffer except containing 300 mM imidazole. The eluate was immediately concentrated and subjected to a Superdex-200 size exclusion column (GE Healthcare) in gel filtration buffer (reduced salt PBS—1.8 mM $KH_2PO_4$, 10 mM $Na_2HPO_4$, 2.7 mM KCl, and 75 mM NaCl). Peaks containing the highest purity protein were collected, and the protein was stored on ice and subjected to ITC as soon as possible.

The binding affinities of polyamines to NspS-6xHis were measured using a MicroCal PEAQ-ITC (Malvern) instrument at 25˚C. Norspermidine and spermidine were each dissolved in the above gel filtration buffer. To measure $K_d$ values, the indicated concentrations of each polyamine were titrated into a solution containing 7 μM apo-NspS-6xHis with continuous stirring

at 750 rpm. The instrument was controlled with PEAQ-ITC Control software (MicroCal), and results were fitted and evaluated by the PEAQ-ITC Analysis software (MicroCal).

## RNA isolation and sequencing

Overnight cultures of the indicated *V. cholerae* strains, grown in biological triplicate, were diluted to $OD_{600}$ approximately 0.001 in 5 mL of fresh M9 medium. These subcultures were grown at 30˚C with shaking in the presence of the designated polyamine and/or arabinose to $OD_{600} = 0.1$. Cells were harvested by centrifugation for 10 min at $3,200 \times g$ and resuspended in RNAprotect (Qiagen). RNA was isolated using the RNeasy mini kit (Qiagen), remaining DNA was digested using the TURBO DNA-free kit (Invitrogen), and the concentration and purity of RNA were measured using a NanoDrop instrument (Thermo). Samples were flash frozen in liquid nitrogen and stored at −80˚C until they were shipped on dry ice to the Microbial Genome Sequencing Center (MIGS; https://www.migscenter.com/rna-sequencing). Upon sample submission, the 12 million paired-end reads option and the intermediate analysis package were selected for each sample. As per the MIGS project report, quality control and adapter trimming were performed with bcl2fastq (Illumina), while read mapping was performed with HISAT2 [40]. Read quantitation was performed using the Subread's featureCounts [41] functionality, and subsequently, counts were loaded into R (R Core Team) and normalized using edgeR's [42] Trimmed Mean of M values (TMM) algorithm. Values were converted to counts per million (cpm), and differential expression analyses were performed using edgeR's Quasi-Linear F-Test (qlfTest) functionality against treatment groups, as indicated. Heatmaps and volcano plots were produced in Python 3 [37].

## Supporting information

**S1 Fig. In vivo stoichiometry of NspS and MbaA proteins and ITC measurements of binding constants.** (A) Relative c-di-GMP reporter output for wild-type *V. cholerae* (left panel) and *V. cholerae* harboring *mbaA-3xFLAG* and *nspS-3xFLAG* (right panel) expressed from the native locus on the chromosome. Treatments: no addition (designated Ctrl), 100 μM spermidine, and 100 μM Nspd. (B) Western blot of MbaA-3xFLAG and NspS-3xFLAG. R1, R2, and R3 designate 3 biological replicates. (C) SDS-PAGE gel showing purity of the NspS-6xHis protein used for ITC measurements. Molecular weight markers are designated on the left. The arrow on the right shows the position of NspS-6xHis. (D) ITC data, plot, and calculated values for Nspd binding to purified NspS-6xHis. (E) As in D for Spd. (F) Shown is the c-di-GMP reporter output for a *V. cholerae* strain carrying *Ptac-nspS-mbaA* at the native *nspS-mbaA* locus. Data are displayed as percent increases relative to the wild-type strain with no polyamines added. Numerical values for plots are available in S1 Data. c-di-GMP, cyclic diguanylate; ITC, isothermal titration calorimetry; Nspd, norspermidine; Spd, spermidine. (TIF)

**S2 Fig. MbaA-3xFLAG levels can be controlled by arabinose and quantified.** Representative western blot (top) and companion quantitation (bottom) of MbaA-3xFLAG levels for *V. cholerae* carrying *mbaA-3xFLAG* at the endogenous *mbaA* locus (referred to as WT, leftmost lane and companion bars) and for the *V. cholerae* Δ*mbaA Pbad-mbaA-3xFLAG* strain following induction by the indicated arabinose concentrations (right three lanes and companion bars). RpoA was used as the loading control in the western blot. Numerical values for plots are available in S1 Data. WT, wild type. (TIFF)

**S3 Fig. Induction of *vps* expression by polyamines requires MbaA diguanylate cyclase activity.** Shown is the *vpsL-lux* reporter output for MbaA carrying the D426A and E427A substitutions. In this mutant MbaA protein, the SGDEF catalytic site is altered to SGAAF, which eliminates c-di-GMP biosynthetic capability. Data are displayed as $\log_2$ FCs relative to the untreated strain (bottom left corner). Numerical values for plots are available in S1 Data. FC, fold change; Nspd, norspermidine; Spd, spermidine; *vps*, vibrio polysaccharide biosynthesis genes. (TIFF)

**S4 Fig. Biofilm gene expression specificity can be achieved by altering the global cytoplasmic c-di-GMP level.** Volcano plots showing FCs in gene expression relative to the corresponding controls, measured by RNA sequencing. FCs were quantified relative to untreated wild-type *V. cholerae* for all conditions except those involving arabinose, which were compared to the wild-type strain treated with the equivalent arabinose concentration. Plots represent the full transcriptional profiles for the strains and conditions shown in Fig 6. Biofilm genes are highlighted in red and motility genes are depicted in orange. The horizontal dotted line represents a *p*-value of 0.05, and left and right vertical dashed lines represent $\log_2$ FCs of −1 and 1, respectively. Samples were collected at $OD_{600}$ = 0.1 and $N$ = 3 biological replicates. Complete datasets are available in S1–S3 Data. c-di-GMP, cyclic diguanylate; FC, fold change; WT, wild type. (TIFF)

**S5 Fig. The c-di-GMP reporter is not saturated in the *V. cholerae vpvc*$^{W240R}$ strain.** c-di-GMP reporter output from *V. cholerae* carrying *vpvC*$^{W240R}$ at the native *vpvC* locus and *Pbad-vpvC*$^{W240R}$ integrated at an ectopic locus without and with the arabinose inducer as indicated. Data are displayed as percent increases compared to the untreated wild-type strain. A Kruskal–Wallis test was performed for statistical analysis. *0.01 < $P$ ≤ 0.05; n.s., $P$ > 0.05. Numerical values for plots are available in S1 Data. c-di-GMP, cyclic diguanylate. (TIFF)

**S6 Fig. The NspS-MbaA system can modify the c-di-GMP pool established by VpvC**$^{W240R}$**.** Experimentally obtained results for c-di-GMP reporter output in *V. cholerae* carrying *Pbad-vpvC*$^{W240R}$ treated with 0.0125% arabinose, for the indicated polyamine concentrations, displayed as a heatmap. Throughout the manuscript, data in c-di-GMP output heatmaps are displayed as percent differences compared to the untreated wild-type strain, with teal representing low and purple representing high c-di-GMP reporter output, respectively. Numerical values for plots are available in S1 Data. c-di-GMP, cyclic diguanylate. (TIF)

**S1 Text. Supplementary text.** (PDF)

**S1 Table. Model parameters.** (PDF)

**S2 Table. Strains used in this study.** (DOCX)

**S3 Table. DNA oligonucleotides and gene fragments used in this study.** (DOCX)

**S1 Data. Numerical values underlying plots in Figs 1–4, 6, S1–S3, S5 and S6.** (XLSX)

**S2 Data. RNA sequencing results for treatment of wild-type *V. cholerae* with 100 μM norspermidine.**
(XLSX)

**S3 Data. RNA sequencing results for treatment of wild-type *V. cholerae* with 100 μM spermidine.**
(XLSX)

**S4 Data. RNA sequencing results for the Δ*mbaA* *V. cholerae* strain.**
(XLSX)

**S5 Data. RNA sequencing results for treatment of the Δ*mbaA* *V. cholerae* strain with 100 μM norspermidine.**
(XLSX)

**S6 Data. RNA sequencing results for treatment of the Δ*mbaA* *V. cholerae* strain with 100 μM spermidine.**
(XLSX)

**S7 Data. RNA sequencing results for the *vpvC*$^{W240R}$ *V. cholerae* strain.**
(XLSX)

**S8 Data. RNA sequencing results for the *V. cholerae* strain carrying *Ptac-nspS-mbaA*.**
(XLSX)

**S9 Data. RNA sequencing results for the *V. cholerae* strain carrying *Pbad-vpvC*$^{W240R}$ following treatment with varying concentrations of arabinose.**
(XLSX)

**S10 Data. RNA sequencing results for the *V. cholerae* strain carrying *Pbad-cdgL* following treatment with varying concentrations of arabinose.**
(XLSX)

**S11 Data. RNA sequencing results for treatment of wild-type *V. cholerae* with 10 μM norspermidine.**
(XLSX)

**S1 Raw Images. Raw gel images for S1 and S2 Figs.**
(PDF)

## Acknowledgments

We thank the members of the Bassler and Wingreen groups for insightful comments and ideas. Isothermal Titration Calorimetry was performed in the Princeton Biophysics Core Facility with Venu Vandavasi.

## Author Contributions

**Conceptualization:** Andrew A. Bridges, Jojo A. Prentice, Chenyi Fei, Ned S. Wingreen, Bonnie L. Bassler.

**Data curation:** Andrew A. Bridges, Jojo A. Prentice, Chenyi Fei, Ned S. Wingreen, Bonnie L. Bassler.

**Formal analysis:** Andrew A. Bridges, Jojo A. Prentice, Chenyi Fei, Ned S. Wingreen, Bonnie L. Bassler.

**Funding acquisition:** Ned S. Wingreen, Bonnie L. Bassler.

**Investigation:** Andrew A. Bridges, Jojo A. Prentice, Bonnie L. Bassler.

**Methodology:** Andrew A. Bridges, Jojo A. Prentice, Chenyi Fei, Ned S. Wingreen.

**Project administration:** Bonnie L. Bassler.

**Resources:** Andrew A. Bridges, Jojo A. Prentice.

**Software:** Jojo A. Prentice, Chenyi Fei, Ned S. Wingreen.

**Supervision:** Ned S. Wingreen, Bonnie L. Bassler.

**Validation:** Andrew A. Bridges, Jojo A. Prentice, Chenyi Fei, Ned S. Wingreen, Bonnie L. Bassler.

**Visualization:** Andrew A. Bridges, Jojo A. Prentice.

**Writing – original draft:** Andrew A. Bridges, Jojo A. Prentice, Chenyi Fei, Ned S. Wingreen, Bonnie L. Bassler.

**Writing – review & editing:** Andrew A. Bridges, Jojo A. Prentice, Chenyi Fei, Ned S. Wingreen, Bonnie L. Bassler.

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
