## [Editor Report · Decision Letter 0]

8 Oct 2021

Dear Dr. Bassler, 

Thank you for submitting your manuscript entitled "Quantitative input-output dynamics for a c-di-GMP signal-transduction cascade in Vibrio cholerae" for consideration as a Research Article by PLOS Biology.

Your manuscript has now been evaluated by the PLOS Biology editorial staff, as well as by an academic editor with relevant expertise, and I am writing to let you know that we would like to send your submission out for external peer review.

Once your full submission is complete, your paper will undergo a series of checks in preparation for peer review. Once your manuscript has passed the checks it will be sent out for review. 

If your manuscript has been previously reviewed at another journal, PLOS Biology is willing to work with those reviews in order to avoid re-starting the process. Submission of the previous reviews is entirely optional and our ability to use them effectively will depend on the willingness of the previous journal to confirm the content of the reports and share the reviewer identities. Please note that we reserve the right to invite additional reviewers if we consider that additional/independent reviewers are needed, although we aim to avoid this as far as possible. In our experience, working with previous reviews does save time. 

If you would like to send your previous reviewer reports to us, please specify this in the cover letter, mentioning the name of the previous journal and the manuscript ID the study was given, and include a point-by-point response to reviewers that details how you have or plan to address the reviewers' concerns. Please contact me at the email that can be found below my signature if you have questions. 

Please re-submit your manuscript within two working days, i.e. by Oct 10 2021 11:59PM.

Kind regards,

Paula

Paula Jauregui, PhD

Associate Editor

PLOS Biology

---

## [Decision Letter · Decision Letter 1]

13 Dec 2021

Dear Dr. Bassler,

Thank you for submitting your manuscript "Quantitative input-output dynamics for a c-di-GMP signal-transduction cascade in Vibrio cholerae" for consideration as a Research Article at PLOS Biology. Your manuscript has been evaluated by the PLOS Biology editors, an Academic Editor with relevant expertise, and by several independent reviewers.

In light of the reviews (below), we will not be able to accept the current version of the manuscript, but we would welcome re-submission of a much-revised version that takes into account the reviewers' comments. We cannot make any decision about publication until we have seen the revised manuscript and your response to the reviewers' comments. Your revised manuscript is also likely to be sent for further evaluation by the reviewers.

In particular, we think it is important that you improve the introduction and discussion to enhance the contextualization of the work, as reviewer #1 suggests. Reviewer #2 questions why the chromosomal vpvC-W240R gene affect transcription of motility genes, but expressing it in trans does not. This reviewer also thinks that you could examine the expression levels of the vpvC-W240R gene in the different backgrounds, show the colony biofilm phenotypes of the expression mutants and whether differential vps-W240R expression explain the different phenotypes, and quantify the absolute levels of c-di-GMP in key strains. Reviewer #3 wants you to test whether in a npsC/potD mutant the variation in level of c-di-GMP is observed, to test whether the response is c-di-GMP-dependent in the wild type, whether NspS bound to spermidine also binds at low frequency to MbaA and how will this be accounted for into the mathematical model, clarify the rationale behind your indication that MbaA unbound to NspS can also display diguanylate cyclase activity, and comment on how the NspS-MbaA system can be adapted to other c-di-GMP signaling systems. Reviewer #4 thinks that the key question would be what the local/pathway-specific effectors that could contribute to physiological polyamine sensing and related MbaA activities are. Please address all the reviewers' issues. 

We expect to receive your revised manuscript within 3 months. 

**IMPORTANT - SUBMITTING YOUR REVISION**

*Re-submission Checklist*

*Published Peer Review*

*PLOS Data Policy*

*Blot and Gel Data Policy*

Sincerely,

Paula

---

Paula Jauregui, PhD

Associate Editor

PLOS Biology

REVIEWS:

Reviewer #1: Bacterial signaling.

Reviewer #2: Bacterial signalling pathways.

Reviewer #3: Bacterial signalling pathways and c-di-GMP.

Reviewer #4: Vibrio and cyclic nucleotides.

Reviewer #1: The manuscript "Quantitative input-output dynamics for a c-di-GMP signal transduction cascade in Vibrio cholerae" describes further characterization of the NspS-MbaA pathway, which regulates biofilm formation and dispersal in response to specific polyamines. This pathway is of particular importance to the field of c-di-GMP signaling and biofilm formation as signals for this pathway have been identified making it possible to study all the steps of the signal transduction cascade from signal input to phenotypic output. The work sheds light on the long-standing issue of how signal specificity may be achieved in signal transduction pathways that use c-di-GMP as a second messenger.

This a beautiful and thorough study that investigates the NspS/MbaA pathway using a powerful and tunable genetic system. It's well written, clear, and concise. The authors build a mathematical model of signal processing by this pathway that accurately explains experimental observations of c-di-GMP levels under different nspd and spd concentrations and changes to MbaA levels. They show that the pathway is very sensitive to nspd with a dissociation constant in the subnanomolar range and 100-fold lower than that for spd. They demonstrate that high affinity import by PotABCD1 leads to very low periplasmic concentrations of nspd and spd, which is consistent with the sensitivity of the NspS/MbaA pathway to these polyamines. They show that this pathway regulates transcription of biofilm-specific genes and not others involved in other c-di-GMP regulated phenotypes such as motility. In addition, using both the NspS/MbaA and other c-di-GMP metabolizing proteins (VpvC(W240R) and CdgL), the authors show that c-di-GMP signaling specificity can be explained by changes in the cytoplasmic c-di-GMP levels and that local signaling is not required for specificity. However, they also demonstrate, local signaling most likely occurs with the NspS/MbaA system since vpsL gene expression is activated at nspd concentrations that do not elevate c-di-GMP pools. 

Major issues

The authors should do a better job of putting current study in the context of previous work by other groups as well. Some of the observations/experiments reported here have been reported by other groups previously. In most of these cases, the techniques used in these earlier studies are different than and not as sensitive as those used in this study and the effects appear to be observed at higher concentrations. However, many of the observations are consistent with those reported in this study and strengthen the claims of the current study. For example, some of the experiments described in fig 3 done with nspC and nspCpotD1 mutants have been reported before (Wotanis et al. 2017) and should be cited. While effect of spd was not measured and only biofilm assays were reported in this study, the response of the nspCpotD1 double mutant to nspd was clearly demonstrated in this publication and is consistent with results in the current study. The idea that "MbaA transmits information internally to elicit gene expression changes at polyamine concentrations below that required to change cytoplasmic c-di-GMP" levels" has also been described before (Sobe et al. 2017) and should be cited. Additionally, effect of nspd on vpsL transcription (Lines 317-318) has been reported (Karatan et. al, 2005). MbaA transcriptomic analysis has also been reported in the same publication and is consistent with the observation that a small set of biofilm-specific genes being affected by the NspS/MbaA pathway (lines 369-372) and should be cited. In cases where results of previous work are not consistent (lack of changes in c-di-GMP pools in response to nspd or in nspS and mbaA mutants, Sobe et al. 2017), an attempt should be made to provide possible explanations for the differences. 

Nevertheless, this should not be perceived as redundancy of the current study with previously published work. On the contrary, the current study does an excellent job of providing explanations to and reconciling many of the observations reported before in the context of nspd/spd signaling through the NspS/MbaA system and the effect of import by PotABCD1 on fine tuning this signaling. 

Minor issues

1. Line 66: Wrong citation. Should be cockerell et al. 2014

2. Line 117: Cockerell et al. 2014 shows the norspermidine import by the PotABCD1 transporter and should also be cited. 

3. Page 8. The free-energy model assumes that one of the three states MbaA can exist as unbound to NspS and exhibiting DGC activity. Given that the nspS mutant makes very low amounts of c-di-GMP and biofilm, it is unclear to me why MbaA unbound to NspS is thought to have DGC activity. Please provide an explanation that will help the audience understand this assumption.

4. Line 255: Please change to "in which nspd production and nspd and spd import" were activated. V. cholerae produces other polyamines (putrescine, diaminopropane, and cadaverine), so the statement as written is inaccurate.

5. Line 262: Please change to "incapable of nspd production" for the reason stated above.

6. Lines 283-284: Please change to "in which nspd production and nspd and spd import".

7. Line 287: Please change to "external nspd and spd".

8. Line 297: Similar to above, please specify the type of polyamine import and export

9. Other places in the manuscript where the type of polyamine needs to be specified: lines 300-301, 519 etc.

10. Line 519: Please provide information on what is considered "physiologically-relevant" polyamine concentrations. 

Reviewer #2: This study uses a clever combination of computational modelling, biochemistry and genetics to dissect the nspS/mbaA cyclic-di-GMP regulatory circuit in V. cholerae. The authors first construct a free-energy model of the relationship between periplasmic polyamine concentrations and the enzymatic activity of a bifunctional cyclic-di-GMP enzyme; MbaA. Earlier, experimentally derived results for cyclic-di-GMP outputs with different inputs of spermidine and norspermidine alongside biochemical determination of key parameters (Kd etc.) were then used to fit this model. The model successfully recapitulated the behaviour of key V. cholerae mutants, and enabled accurate predictions of states where MbaA / polyamine abundance was perturbed. The authors showed that the NspS-MbaA circuit is exceptionally sensitive to changes in periplasmic levels of norspermidine, and this manifests in downstream cyclic-di-GMP-mediated changes in bacterial behaviour. Polyamine signalling was shown to work exclusively through MbaA, and cyclic-di-GMP produced by MbaA was shown to exclusively control transcription of biofilm genes. The authors go on to test the principles of the global and specific cyclic-di-GMP signalling models, using mbaA alongside other Vibrio DGCs to modulate cyclic-di-GMP and gene transcription. They present evidence that both models probably function in V. cholerae. I have a few comments on the manuscript as it stands:

Major comment

Fig 6: Why does the chromosomal vpvC-W240R gene affect transcription of motility genes, but expressing it in trans to a level that apparently produces the same amount of c-di-GMP does not? On Line 462 the authors state the dynamic range of their system is not enough to achieve the same results seen for the chromosomal mutation, but this seems surprising. DGC genes are not generally expressed at high level, and it seems likely the plasmid borne copy is expressed at higher levels than the chromosomal mutant. Looking at the 1st and 11th columns of the chart in fig 6, the cyclic-di-GMP levels look pretty similar to me. Are these values significantly different from one another? 

I suspect that the c-di-GMP measurements seen in the vpvC-W240R mutant and mbaA over-expression strains might be saturating, and do not reflect the true levels of the molecule in these strains. This would explain why vpsL needed to be deleted in these two backgrounds only. If this is the case, then it is difficult to compare the results from these two strains with the results seen for the other strains in this experiment. 

This raises a few questions: what is going on with the vpvC-W240R over-expression strain? Did the authors see similar aggregation here? Is this gene really expressed at a lower level in these strains, or is something else, e.g. another, unidentified mutation in the chromosome of vpvC-W240R the cause of these discrepancies? 

This could be cleared up by examining the expression levels of the vpvC-W240R gene in these backgrounds by qRT-PCR [or possibly from the RNA seq data]. Likewise, the colony biofilm phenotypes of the expression mutants would be useful to see here or in a supplementary figure. If differential vps-W240R expression cannot explain the different phenotypes seen here, then the authors need to work out what is going on in this strain, possibly by sequencing it and looking for other compensatory mutations that may enhance cyclic-di-GMP levels. Finally, if the c-di-GMP assay is saturating, then the authors should quantify the absolute levels of c-di-GMP in key strains using LC/MS. 

Minor points:

1. Line 41: Ensure the VPS abbreviation is explicitly defined here.

2. Line 82 onwards: I don't think this can be stated as an either/or question. There is strong evidence for both delocalized effector affinity [https://onlinelibrary.wiley.com/doi/full/10.1111/mmi.12066] and localized signalling networks [https://journals.asm.org/doi/10.1128/mBio.01639-17] operating in different bacterial contexts. The authors should consider rephrasing this section more towards an assessment of the relative importance and potential overlap of these different mechanisms, rather than presenting a binary choice between them. 

3. Figure 5E: The legend description for this panel is rather ambiguous and should be clarified. What does this plot represent? I guess this is mutant vs WT?

4. Line 389-408 and Fig 5E: Point mutants in other DGC enzymes (e.g. WspR19 in Pseudomonas, PleD* in Caulobacter) have been shown to induce overproduction of c-di-GMP far in excess of the physiological maximum for the system in nature, typically by disabling product inhibition. The authors need to show evidence that this is, if not impossible here, then at least unlikely. The authors should state what is known about the activation mechanism of the W240R mutation, from the earlier work of Beyhan and Yildiz.

5. Line 409-428: The extensive description of these two signalling models would be better placed in the discussion. The models could be briefly introduced here for the purposes of continuity with the following section, but then discussed more thoroughly later. 

6. Fig 6: The scale on the bottom panel doesn't make sense to me. If these values are relative to WT, then how can they be expressed as both a positive and a negative percentage of reporter output? 

Reviewer #3: This is a very interesting manuscript addressing a pending question about how the information lying into the universal second messenger c-di-GMP is able to be transduced into multiple and specific responses in bacteria. Here the authors present a combination of experimental and modelling data, using one of many diguanylate cyclases of Vibrio cholerae, MbaA, and testing how the response to polyamine, via NsbS-MbaA interaction, is effectively transduced, locally or globally and with a general or specific impact on Vibrio physiology, notably biofilm formation.

Although I have no appropriate expertise to assess the buildup of the mathematical modeling, the biology presented holds a number of novel concepts, or provides previously accepted concepts with experimental validation here. In general, the paper is rather dense and somehow complex, although the concepts that are conveyed are straightforward.

Here below are a few comments that mostly relates to the biological aspect of the work:

- The authors showed that in absence of production (nspC mutant) and import of polyamine (potD mutant) the detection of exogenously added norspermidine is highly sensitive (sub-nanomolar range) as monitored by biofilm formation (line 308). Biofilm is also driven through production of the VPS polysaccharide. Subsequently the authors test the reporter vpsL-lux against the addition of polyamine to wild-type Vibrio and observed that addition of micromolar range of norspermidine has no effect on global c-di-GMP but does impact biofilm and vpsL-lux expression. This led the authors to conclude to a local mechanism of transmission of c-di-GMP signaling, whose variation in concentration is not seen at the global level, but still effects a specific response on VPS. i) It would be appropriate to test whether in a npsC/potD mutant, and not a wild-type, the variation in level of c-di-GMP is observed, and that cannot be seen in the wild-type due to depletion of the periplasmic norspermidine upon its transport into the cytoplasm. ii) It would also be appropriate to test whether the response is c-di-GMP-dependent in the wild type by using a mutant in which the GGDEF motif of MbaA has been mutated so that there is no longer diguanylate cyclase activity. This way it confirms that the despite the lack of global change in c-di-GMP it is a c-di-GMP-dependent response that is observed.

- One assumption that comes out of the work (line 544) is that a small fraction of NsbD unbound to norspermidine (apo-NspS) would interact with MbaA. Could the authors discuss whether this reflects a difference in affinity between the close and open state of NspS. Would it be possible that NspS bound to spermidine also binds at low frequency to MbaA and how will this be accounted for into the mathematical model?

- Lines 159-160 it is indicated that MbaA unbound to NspS can also display diguanylate cyclase activity. I am not sure to clearly grasp what is the rationale behind this. Could the authors clarify? Does purified MbaA have cyclase or phosphodiesterase activity? Which activity, cyclase or phosphodiesterase, does a truncated MbaA carrying only the GGDEF and EAL domain have?

- It would be appropriate to briefly comment on how the NspS-MbaA system can be adapted to other c-di-GMP signaling systems as mentioned on line 565.

Reviewer #4: In this submission, Bridges, Bassler and colleagues continue their investigation in the polyamine-sensing system controlling intracellular c-di-GMP in Vibrio cholerae, whose key components are the NspS polyamine periplasmic receptor and its binding partner, the inner-membrane bifunctional diguanylate cyclase/phosphodiesterase enzyme MbaA. In contrast to other inner-membrane, ligand-sensing c-di-GMP regulatory systems, the ligands for the NspS/MbaA partners have been identified and depending on the specific bound polyamine and related ligand- and protein partner-binding affinities, the system can switch from c-di-GMP generation to c-di-GMP degradation with the associated inverse effects on biofilm formation and dispersal, respectively. 

The manuscript is clearly written and easy to follow and the data representation, as typical for this group's works, is neat and self-explanatory. The experiments are well controlled and logical and overall of good quality. The authors present simple and intuitive mathematical models that however satisfactorily describe the observed effects of ligand modulation on c-di-GMP levels and biofilm formation. If anything, the text can profit from a better introduction of the role of spermidine and norspermidine on the pathogen's physiology and in particular in relation with biofilm formation vs. dispersal in the environment and the host. The physiological concentration ranges for the two polyamines, if known, would be also very relevant to refer to throughout the study.

My major concern about this article is that it is quite incremental with regard to recent findings by the same group published elsewhere and in particular the Bridges & Bassler eLife paper from a few months ago where the system's workings were reported with regard to biofilm dispersal, a process intrinsically inverse to biofilm formation. The mechanism and effects of norspermidine vs spermidine sensing via NspS, MbaA and even associated partners involved in polyamine import were already beautifully reported in that study. From the underlying hypotheses, to the specific experimental toolkit and examined mutants, the current manuscript is mostly an intuitive continuation of the previous study. While the current submission provides a more quantitative rationalization of the observed effects, overall I don't find it provides substantial new insights into the mechanisms of Vibrio biofilm formation. 

What seemed potentially interesting is the possibility to distinguish between local signal transduction through direct generation-sensing-degradation of c-di-GMP among interacting or spatially constrained proteins vs. modulation of the global pool of available c-di-GMP. Overall, however, I feel that the manuscript falls short in doing that. 

If I am not wrong many, if not most, c-di-GMP signaling systems experience some degree of pathway specificity which is in line with the presence of multiple non-redundant DGCs, PDEs or bifunctional enzymes per genome, including that of V. cholerae. For example, work from the Hengge lab and others have shown that secretion of biofilm matrix components such as polysaccharides and curli in enterobacteria (e.g. E. coli and Salmonella) is often controlled by a multilevel cascade of c-di-GMP sensing proteins and interacting DGCs, PDEs and even proteins directly involved in secretion. Yet, almost any active DGC induced in a standard E. coli protein expression strain that barely secretes extracellular polymers can lead to secretion of biofilm matrix components circumventing both pathway-specific enzymes and spatial restrictions (which is actually routinely used as an assay for identification of enzymatically active DGCs). The point being that the biofilm-stimulating effects observed by non-specific modification of global c-di-GMP levels in Vibrio here, even upon adjusting for cellular c-di-GMP concentrations with regard to the NspS/MbaA system are not particularly surprising, especially since the read-out of cytosolic c-di-GMP concentrations is itself determined by c-di-GMP complexation by the used reporter. 

Interestingly, the authors do observe some specific effects at low and likely more physiological polyamine concentrations where the global c-di-GMP concentration does not exhibit significant changes (again, dependent on the used reporter). 

The key question, and answer, here that can bring substantial novelty to the work and make it suitable for publication in PLoS Biology would be what are the local/pathway-specific effectors that could contribute to physiological polyamine sensing and related MbaA activities? The authors offer several specific leads in the discussion yet none of these were experimentally tested and overall the results and discussion related to the local vs global c-di-GMP sensing remain mostly handwaving. For example, VpsT is known to complex c-di-GMP and form supramolecular clusters that can be somewhat easily observed in spermidine vs norspermidine exposure and may be co-localized with the proteins in question, even if high-affinity interactions do not necessarily occur stably. Protocols for expression (and purification) of both VpsT and VpsR, two biofilm transcriptional regulators and potential MsbA partners discussed by the authors are available and could be tested for interactions with the system's components both in cell-based and in vitro assays that seem well within the expertise of the group. Of course, an unbiased protein partner screening approach would be preferred as biofilm-promoting effectors are not necessarily part of the known transcription regulators.

Somewhat minor comments: it seems the effects on biofilm-stimulating genes is more pronounced than those on flagellar motility genes even in the context of very high cytosolic c-di-GMP, so maybe it is not surprising that the biofilm genes are the first to be detected upon subtle changes in the polyamine concentrations/MbaA activity changes. As the authors only looked at transcriptional changes, whereas local signaling effects can be also exerted on already expressed effectors (i.e. post-translationally), differential effects would be also dependent on the target promoters and not only on the c-di-GMP-binding affinities of the involved transcriptional regulators. These caveats should be considered in the models and discussion. 

In conclusion, this article provides additional examination of a system already quite well characterized by the Bassler group at the level of spermidine/norspermidine sensing and the inverse effects of the two polyamines on MbaA activity and downstream biofilm effects. It pains me to write this, but I do not believe the novelty and insights gained here are nearly sufficient for a publication in PLoS Biology or another novelty-driven PLoS journal unless further data on the downstream signal effectors are indeed reported.

---

## [Editor Report · Decision Letter 2]

25 Feb 2022

Dear Dr Bassler,

On behalf of my colleagues and the Academic Editor, Matt Waldor, I am pleased to say that we can in principle accept your Research Article "Quantitative input-output dynamics of a c-di-GMP signal-transduction cascade in Vibrio cholerae" for publication in PLOS Biology, provided you address any remaining formatting and reporting issues. These will be detailed in an email that will follow this letter and that you will usually receive within 2-3 business days, during which time no action is required from you. Please note that we will not be able to formally accept your manuscript and schedule it for publication until you have addressed any requested changes.

IMPORTANT: Many thanks for providing the data underling all main figures and supplementary ones. I've asked my colleagues to also request that you include information about the underlying data location in each figure legend (including in the supplementary figure legends). I've also asked to request that you provide the original image supporting the results reported in Supplementary Figure 1B. 

PRESS

Thank you again for choosing PLOS Biology for publication and supporting Open Access publishing.

We look forward to publishing your study. 

Sincerely, 

Dario

Dario Ummarino, PhD 

Senior Editor 

PLOS Biology

dummarino@plos.org